# Tuning valley polarization of moiré trapped biexcitons by fine structure occupation in WS$_2$/WSe$_2$ heterostructures

Yufei Jiang[1,5], Yongzhi She[1,5], Xinke Cheng[1], Qinghai Tan [2], Jinlong Yang[1], Yilong Zhao[1], Peiwu Liu[1], Min Wu[1], Xiaotian Dai[3], Zengkai Wang[3], Hongbing Cai [3] ✉, Nan Pan [3,4] ✉ & Xiaoping Wang[1,3,4]

Two-dimensional transition metal dichalcogenides (TMDs) heterostructures formed moiré superlattices have emerged as a new platform for exploring correlated excitonic states and valleytronic phenomena. Despite the significant progress in moiré-trapped single excitons, the valley polarization and fine structure of moiré-trapped biexcitons remain poorly understood, with the existing studies reporting only limited or zero polarization and lacking insight into the underlying mechanisms. Here, we study the moiré-trapped interlayer biexcitons in WS$_2$/WSe$_2$ heterostructures through power- and temperature-dependent photoluminescence (PL) spectroscopy. We find that the valley polarization of these biexcitons can be effectively tuned, reaching ~55% at 120 K. This behavior is attributed to the different occupation of intravalley and intervalley biexcitons within the fine structure, with the intravalley biexcitons playing a dominant role. The power-dependent energy splitting and temperature-dependent polarization trends further confirm the existence of biexciton fine structure and its influence on valley polarization. Furthermore, the experiment revealed a fine structure splitting of 2.77 meV, consistent with theoretical calculations. Our study provides new insight into the rational control of excitonic states in moiré superlattices and establishes a basis for developing advanced valleytronic devices, such as polarization-sensitive photodetectors and quantum light sources.

Vertical heterostructures of monolayer transition metal dichalcogenides (TMDs) with type-II band alignment host interlayer excitons (IX) possessing out-of-plane electric dipole moments[1–3]. The presence of a moiré superlattice, arising from twist angles or lattice mismatch, imposes a periodic potential landscape that strongly influences the behavior of IX[4]. Crucially, this platform reveals a rich hierarchy of strongly correlated quantum phases. When the exciton density is lower than the moiré density, the moiré potential traps individual excitons, forming moiré-trapped interlayer single excitons (IX)[5]. As exciton density matches the moiré density, strong dipolar repulsion drives the formation of an incompressible Mott insulator state[6], manifesting as ordered quantum emitter arrays[4]. A significant transition occurs when the exciton density surpasses the moiré density: the system stabilizes novel quantum bound states—moiré-trapped biexcitons (IXX)—where two excitons are confined within a single moiré potential (Fig. 1a)[6–8]. This confinement is critical, as it overcomes the limitation of biexciton dissociation due to repulsive dipole interactions, enabling stable biexciton existence within the moiré potential[9]. Furthermore, the

[1]School of Physical Sciences, University of Science and Technology of China, Hefei, China. [2]School of Microelectronics, University of Science and Technology of China, Hefei, China. [3]Hefei National Research Center for Physical Sciences at the Microscale, University of Science and Technology of China, Hefei, China. [4]Hefei National Laboratory, Hefei, China. [5]These authors contributed equally: Yufei Jiang, Yongzhi She. ✉e-mail: coldice@ustc.edu.cn; npan@ustc.edu.cn

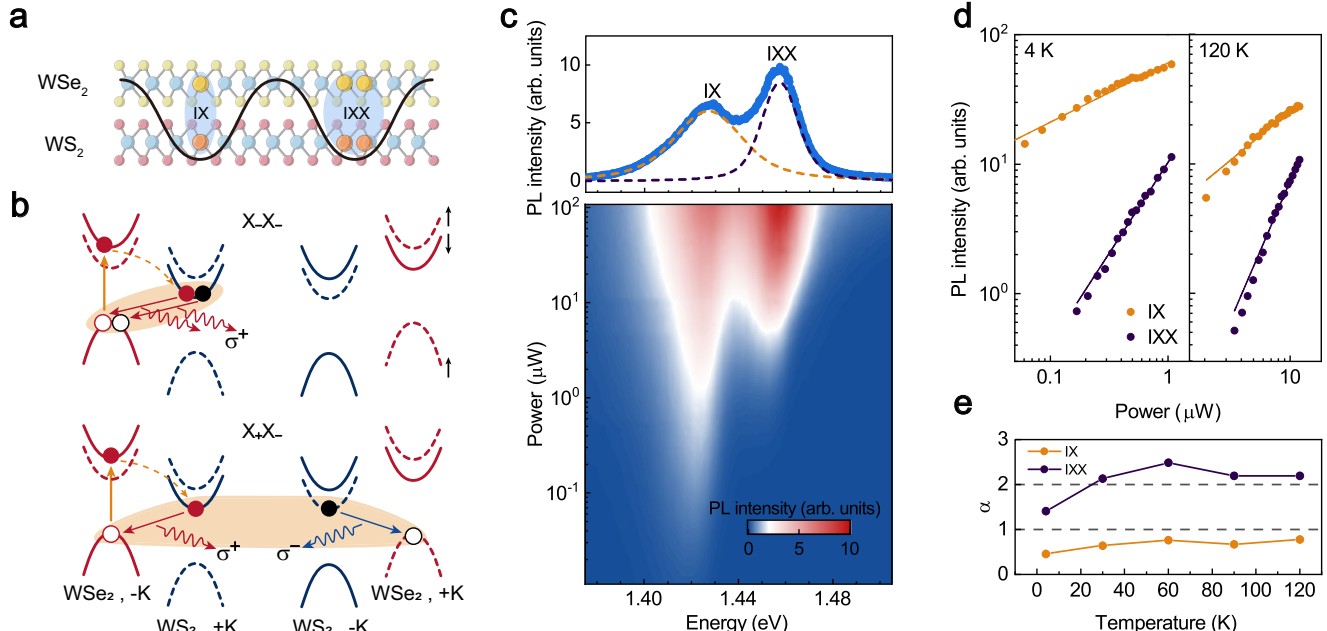

**Fig. 1 | Valley configuration and power-dependent properties of moiré-trapped biexciton. a** Schematically illustrating moiré-trapped interlayer single exciton (IX) and biexciton (IXX). The electrons and holes are localized in the $WS_2$ and $WSe_2$ layer, respectively, resulting in the formation of interlayer excitons. The black curve represents the moiré potential, where a single exciton is confined by the moiré potential to form the IX, while two excitons are confined by the moiré potential to form the IXX. **b** Two valley configurations of IXXs. The lower is intervalley biexciton $X_+X_-$, which is composed of two paired excitons from two different valley, which can emit $\sigma^+$ and $\sigma^-$ light, so it is unpolarized. The upper is intravalley biexciton $X_-X_-$, which can emit circularly polarized light $\sigma^+$. The yellow solid line represents the excitation of the $-K$ valley in $WSe_2$, while the yellow dashed line represents

interlayer charge transfer. Red solid and dashed curves, and blue solid and dashed curves, represent spin-down and spin-up, respectively. **c** Power-dependent PL spectrum (bottom) and representative spectral profile of interlayer exciton at $108\,\mu W$ (the top panel, the PL spectra was fitted using a double Voigt model), a 1.68 eV pulsed laser was used to excite intralayer exciton in $WSe_2$. **d** Extracted PL intensity of IX and IXX as a function of excitation power at 4 K and 120 K. We fit the PL intensity with the power law $I \propto P^\alpha$, PL intensity of IX grows sublinearly ($\alpha = 0.46$ at 4 K and $\alpha = 0.78$ at 120 K), while PL intensity of IXX grows superlinearly ($\alpha = 1.40$ at 4 K and $\alpha = 2.20$ at 120 K) as power increases. **e** Temperature-dependent the power-law exponent $\alpha$ of IX and IXX, where the excitation power is above the threshold power. The dashed line is a guide to the eye.

moiré trapping effect profoundly alters the biexciton binding energy. Dipole repulsion causes the energy of moiré-trapped biexcitons to be higher than that of single excitons[10], which is completely different from the situation in monolayer systems where biexciton energy is lower than that of single exciton[11,12], and fundamentally modifies the properties of biexcitons.

This unique stabilization of repulsive biexcitons unlocks remarkable valleytronic phenomena. In monolayer TMDs, only intervalley biexcitons exist (stabilized by attractive exchange), while intravalley biexcitons are unstable due to intravalley repulsive interaction[13]. For example, in W-based monolayer TMDs, an intervalley biexciton is composed of a bright exciton and a spin-forbidden dark exciton from the opposite valley[14,15]. Thus, the optical selection rule is determined by bright exciton[16]. Whereas in Mo-based monolayer TMDs, the biexciton is composed of two bright excitons from the opposite valley[17,18] typically exhibiting negligible PL polarization. In contrast, the moiré-trapped system stabilizes both intervalley and intravalley IXX bound by moiré potential. This valley-configuration diversity manifests in distinct optical signatures (Fig. 1b): intravalley IXX emits circularly polarized light, while intervalley IXX emits unpolarized light. This inherent valley addressability and polarization control offer the potential for novel valleytronic devices[19,20]. Furthermore, this unique platform bridges directly to quantum information. For instance, the specific valley configurations of moiré-trapped IXX and their cascaded emission[12] pathways hold potential for generating polarization-entangled photon pairs, with intravalley and intervalley IXX offering distinct entanglement states. Consequently, moiré-trapped biexciton systems have emerged as a promising platform for quantum simulation and photonic quantum technologies. However, despite their

fundamental importance and technological promise, critical issues of moiré-trapped biexcitons remain poorly understood. In fact, although a few reports have observed moiré-trapped biexcitons in TMDs heterostructures, only zero[7] or weak[21] valley polarization could be found. The fine structure—specifically, the energy level splitting induced by exchange interactions, which results in distinct energies for intravalley and intervalley biexcitons—and its influence on the valley polarization remains unclear and requires further investigation.

In this work, we identified moiré-trapped IXX and its fine structure in the h-BN encapsulated $59.8° \pm 0.2°$ $WS_2/WSe_2$ heterostructure. We observed that both the valley polarization and the helicity-dependent energy splitting of the IXX increase with the excitation power. This is attributed to the influence of excitation power on the occupation change between intervalley and intravalley IXXs, with the unusually high valley polarization dominated by the intravalley IXX. The behavior of biexcitons can be well understood by the spin-1/2 Bose-Hubbard model. Moreover, the above explanation of the occupation variation of IXX is further confirmed independently by measurements of the temperature dependence of valley polarization. Furthermore, the experiment revealed a fine structure splitting of 2.77 meV, consistent with theoretical calculations. These results not only enhance the understanding of moiré-trapped biexcitons but also provide new insights into the rational control of excitonic states in moiré superlattices, which establishes a basis for developing advanced valleytronic devices.

## Results and discussion
### Interlayer biexcitons in $WS_2/WSe_2$ heterostructure
Figure 1c shows the typical power-dependent interlayer exciton PL spectra in the H-stacked $WS_2/WSe_2$ heterostructure at 4 K (the device's

optical image and SHG signals are shown in Supplementary Section 1). In the Supplementary Section 2, the intralayer exciton luminescence of monolayer and heterostructure regions are dominated by neutral excitons rather than charged excitons, indicating the electrical neutrality of the sample. All measurements were conducted using a laser energy of 1.68 eV to resonantly excite the intralayer excitons in WSe$_2$, unless specified otherwise. At low excitation powers, the interlayer exciton PL spectra only exhibit a single peak with an energy of 1.42 eV. As the power increases, a new peak appears on the higher energy side at approximately 1.45 eV. The upper section of Fig. 1c shows the PL spectrum at high excitation power. The yellow and purple dashed lines represent the fitting of two peaks using a double-Voigt function, clearly identifying both peaks. Notably, as the excitation power increases, the peak positions of the two emission peaks blueshift accordingly (as shown in Supplementary Section 3), and the energy difference between these two peaks remains nearly constant at approximately 30 meV, regardless of the excitation power (see Supplementary Section 3). This observation is consistent well with the recent studies suggesting that the low-energy and the high-energy peaks originate from moiré-trapped IX and IXX[6,7]. The moiré-trapped IXX appears when the excitation power exceeds the threshold of 0.2 μW, exhibiting a significant threshold effect. The exciton density is estimated to be approximately 10$^{12}$ cm$^{-2}$ at this threshold power, nicely comparable to the moiré density of 1.8 × 10$^{12}$ cm$^{-2}$ (see Supplementary Section 4). This threshold aligns well with the characteristic of the moiré-trapped biexcitons within Bose-Hubbard model[6]. Additionally, this threshold behavior of IXX peak further rules out the possibility of the excited states, which can be observed at very low power[22]. To rule out the involvement of other moiré states, we measured the g-factors of IX and IXX (Supplementary Section 5), both of which are found close to +14. This indicates that both IX and IXX originates from the emission of spin-singlet excitons. These results aligns with the observed phenomenon that IX confined in the $H_h^h$ minima, with the excitons in −K (+K) valley emit $\sigma^+$ ($\sigma^-$) light[23], as shown in Fig. 1b. We also ruled out several other possible origins of IXX such as quantum emitters and charged excitons, based on the interlayer and intralayer excitons' PL spectral analyses, as detailed in the Supplementary Section 5.

The exciton types of the IX and IXX can be further identified through power-dependent PL intensity. As discussed in Supplementary Section 6, the power-dependent characteristics of the IX and the IXX at high powers deviate from the low-power behavior due to the filling effects in the moiré superlattice. Therefore, we analyze their intensity's growth near the threshold powers. Figure 1d illustrates the PL intensities of IX and IXX as a function of excitation power at temperatures of 4 K and 120 K. The PL intensity of IX increases sub-linearly with power, whereas that of IXX exhibits a super-linear power dependence. The broadening of the emission peaks of IX and IXX has only a weak influence on the power-law exponent (as discussed in Supplementary Section 6). We then measure the power-law exponent $\alpha$ for both IX and IXX over the temperature range from 4 K to 120 K (The data at 30 K, 60 K and 90 K are displayed in Supplementary Section 6). One can find that both exponents generally increase with the rising temperature and stabilize near $\alpha$ = 1 for IX and $\alpha$ = 2 for IXX (Fig. 1e) consistent with the ideal values for the single excitons and biexcitons, respectively[24,25]. The increase in the power-law exponent with the rising temperature has also been reported in other studies[24,26], and we speculate that it may be related to the thermal activation effects of the excitons trapped by shallow non-radiative potential. More details can be found in Supplementary Section 6.

Furthermore, the moiré-trapped biexciton is of particular interest due to possible cascade emission from the biexciton to the exciton states and from the exciton to the ground states. As shown in Supplementary Section 7, we measured the time-resolved photoluminescence of IXX and IX and found that the decay of IXX is obviously accompanied with an injection process into IX, which

provides clear evidence for the cascade emission of IXX. Based on the experimental observations, we also established a rate equation for the IXX and the IX. We found that considering the diffusion of IX, the fitting results are in good agreement with the measured power-dependent PL intensities (Supplementary Section 7). This phenomenon warrants more in-depth studies in the future.

**Power-dependent valley polarization of the biexcitons**

Figure 1b as well as Supplementary Section 8 reveals the distinct valley configurations in biexcitons, prompting us to investigate the influence of their valley structures on valley polarization. Figure 2a shows the $\sigma^\pm$ circularly polarized PL spectra at 4 K, a $\sigma^-$-polarized light is applied to selectively excite the −K valley at various powers (The excitation modes are indicated in the figure). From the spectra, one can clearly find that the excitation power can effectively tune the valley polarization of both IX and IXX. Figure 2b presents variation of the degree of circular polarization (DCP) as a function of the circularly polarized excitation power. Here the DCP is defined as DCP = $\frac{I_{co}-I_{cross}}{I_{co}+I_{cross}}$, where $I_{co}$ and $I_{cross}$ are co-polarized ($\sigma^-$) and cross-polarized ($\sigma^+$) PL intensities, respectively. These intensities are obtained by fitting the interlayer exciton doublet with a double-Voigt function. At low powers, the IXX is completely absent, the IX dominates the PL emission with a near-zero slightly negative valley polarization. This is consistent with the weakly negative valley polarization observed in the H-type WS$_2$/WSe$_2$ heterostructure[27–29]. When the power slightly exceeds the threshold power, the IXX emission emerges and meanwhile the negative valley polarization of IX begins to increase. As the power further increases beyond 10 μW, the IX valley polarization saturates at about −25%.

It is particularly interesting to find that the valley polarization of the IXX is in sharp contrast to that of the IX. As depicted in Fig. 1b, the intravalley biexcitons can emit circularly polarized light, while the intervalley biexcitons only emit unpolarized light. When the excitation power slightly exceeds the threshold power $P_{th}$ of 0.2 μW, the IXX emerges but hardly have the valley polarization, implying the unpolarized X$_+$X$_-$ configuration of IXX (the middle panel in Fig. 2c)[10,30]. However, as the power further increases, the negative valley polarization of IXX rises and reaches −35% at $P$ = 100 μW. This strongly suggests the generation of intravalley biexcitons X$_-$X$_-$. To eliminate the influence of pulsed laser excitation, we also measured the polarization-resolved spectra under continuous wave laser excitation, which yielded consistent results as shown in the Supplementary Section 9.

The behavior of biexcitons can be well described by the Spin-1/2 Bose-Hubbard model[30]. Within this model, the different valley distributions of biexcitons need to be considered:

$$H = -t \sum_{ij\alpha} b_{i,\alpha}^\dagger b_{j,\alpha} + h.c. + U_{intra} \sum_{i\alpha} n_{i,\alpha}(n_{i,\alpha}-1) + U_{inter} \sum_i n_{i,1} n_{i,2}$$
$$+ V_{DD} \sum_{<i,j>} n_i n_j R_{ij}^{-3}$$

$$(1)$$

Here the $\alpha = \pm1/2$ label +K and −K valley, $t$ is tunning energies between different sites, the $U_{intra}$ and $U_{inter}$ represent the onsite interactions within the intravalley and intervalley IXX. Notably, the excitons in TMDs can be described as spinor boson[10,30,31], where "spin" represents valley pseudospin. The exchange interaction of spinor boson causes the intravalley IXX has a higher energy than the intervalley IXX[7,10] ($U_{intra} > U_{inter}$). As illustrated in Fig. 2c, at low excitation powers (as indicated by the yellow harmonic oscillator well in the left panel in Fig. 2c), the IXX is completely absent and the exciton densities of X$_-$ slightly exceed that of X$_+$ due to its weak valley polarization (X$_-$ and X$_+$ represent exciton in −K and +K valley, respectively). When the excitation power slightly exceeds the threshold power $P_{th}$, the low-energy X$_+$X$_-$ band starts to be occupied and dominate the

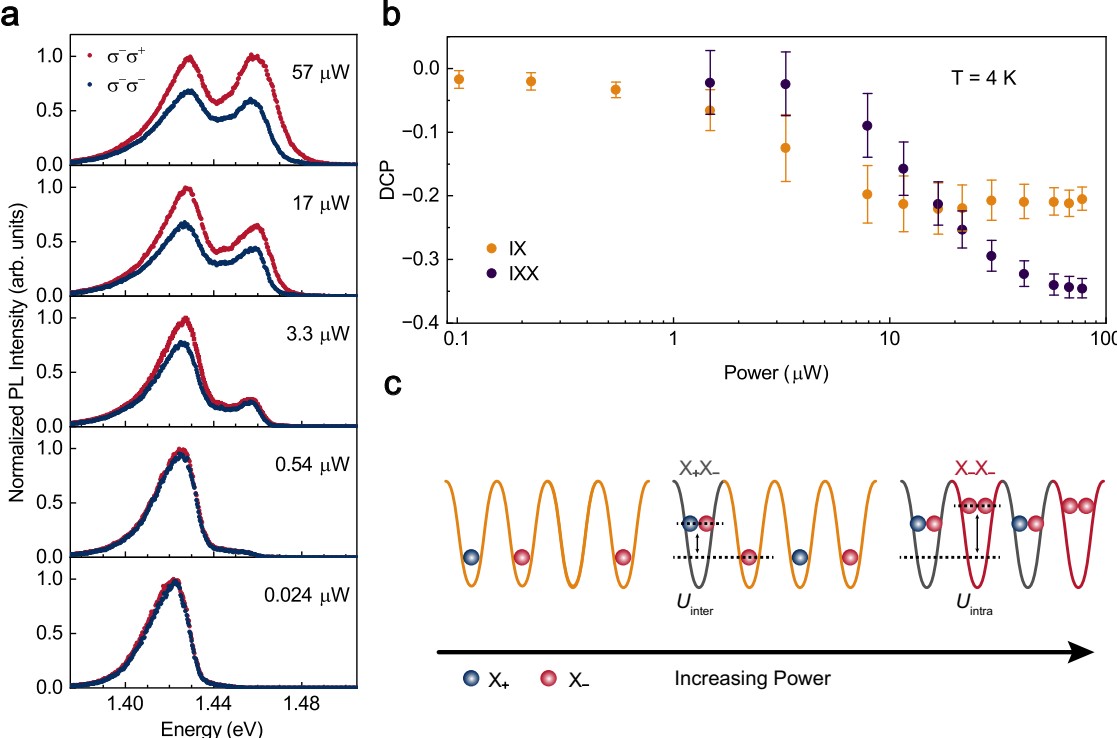

**Fig. 2 | Power-dependent polarization of IX and IXX at 4 K. a** Circular polarization-resolved interlayer exciton PL spectra at selected $\sigma^-$ polarized excitation power. Both IX and IXX are negative valley polarization at high excitation power. **b** DCP of IX and IXX as function of excitation power. Error bars are derived from the fitting uncertainty of the PL intensity of two circularly polarized emissions. **c** Schematic of interlayer exciton filling in moiré superlattices. Blue and red balls represent the excitons in the +K and −K valleys. At low excitation powers (left), only the interlayer single excitons are excited. As power increases, $X_+X_-$ forms in the moiré sites each filled by a +K valley exciton (middle). At high power intensities, both $X_-X_-$ and $X_+X_-$ are excited (right). The $U_{intra}$ and $U_{inter}$ represent the onsite Coulomb interactions for intravalley and intervalley IXX. The red, gray, and yellow harmonic oscillator wells are occupied by the intravalley IXX, intervalley IXX, and IX, respectively.

emission (as indicated by the gray harmonic oscillator well in the middle panel in Fig. 2c). Therefore, the biexciton valley polarization is zero. However, at high excitation powers, the high-energy $X_-X_-$ band also becomes occupied (see the red harmonic oscillator well in the right panel in Fig. 2c), leading to remarkable valley polarization. It is worth to noting that, since the energy difference between the IXX and IX is due to the Coulomb repulsion, the IXX energy level exists only in the moiré potential wells occupied by IXX, just as the IX energy level exists only in the moiré potential wells occupied by IX. We detail the intra- and inter-valley relaxation processes from IXX to IX states in Supplementary Section 10.

Furthermore, the occupation of $X_-X_-$ can also be observed in the PL lifetime. As shown in Supplementary Section 11, the valley polarization is related to the initial valley polarization $P_0$ and the ratio of the exciton lifetime $\tau$ to the valley lifetime $\tau_v$[32], which can be expressed by DCP = $\frac{P_0}{1+2\tau/\tau_v}$. With the increasing excitation power, we observe a significant increase in $P_0$ from near zero to a large negative value (see Supplementary Section 11), indicating a remarkably increased contribution of the intravalley $X_-X_-$ (whereas $P_0 = 0$ for intervalley biexciton $X_+X_-$).

The mechanism can also explain the behavior of increasing valley polarization of the IX at high powers. Owing to the strong and fast intervalley scattering, the exciton densities of $X_-$ and $X_+$ approach equality at low powers, resulting in the valley polarization of IX remaining near zero (the left panel in Fig. 2c). When the power exceeds the threshold for the IXX formation, the excited $X_-$ tends to combine with an $X_+$ to form an $X_+X_-$ intervalley biexciton (the middle panel in Fig. 2c), as shown in Supplementary Section 12. The recombination of the $X_+X_-$ will decay into $X_+$ and $X_-$ with equal probability. Such a process simultaneously consumes and reduces the density of $X_+$,

thereby also enhancing the valley polarization of IX. However, at even higher powers, the formation of $X_-X_-$ consumes an $X_-$ but also leads to the release of an $X_-$. This results in no net increase in the density of $X_-$, and naturally does not enhance the valley polarization of the IX. This dual effect—enhancement at intermediate powers and weakening at high powers—provides the better understanding of the power-dependent valley polarization dynamics.

**Fine structure of interlayer biexcitons**

The existence of the intravalley IXX can also be corroborated by the emission peak positions. Figure 3a shows the energy level diagram of the IXX. At low excitation powers, the lower $X_+X_-$ band is occupied, whereas at high excitation powers, the occupation of the higher $X_-X_-$ band also increases. Considering the circularly polarized emission, $\sigma^+$ emission of biexcitons is contributed by both the $X_-X_-$ and the $X_+X_-$, whereas $\sigma^-$ emission originates primarily from the $X_+X_-$. As a result, when the occupation of the intravalley $X_-X_-$ increases, the peak of $\sigma^+$ PL emission should shift to a higher energy than that of $\sigma^-$. To verify this, we analyze the normalized helicity-resolved PL spectra of the biexcitons (as shown in Fig. 3b), under the $\sigma^-$-polarized excitation at selected powers. We find that, at low excitation power, the peak energy of the $\sigma^\pm$ emission remains the same, indicating the $X_+X_-$ dominant regime. However, in contrast, as the power increases, the peak position of $\sigma^+$ emission shifts notably to a higher energy than that of $\sigma^-$, clearly demonstrating the contribution of $X_-X_-$. By subtracting the trivial influence of IX, we can fit the peak energy of $\sigma^-$ and $\sigma^+$ emission and hence obtain their energy splitting (whose potential origin is discussed in detail in Supplementary Section 13). Figure 3c plots the $\sigma^\pm$ PL peak energy splitting of the IXX as a function of the excitation power under both $\sigma^-$ and $\sigma^+$ excitation, respectively. Here the splitting is defined as

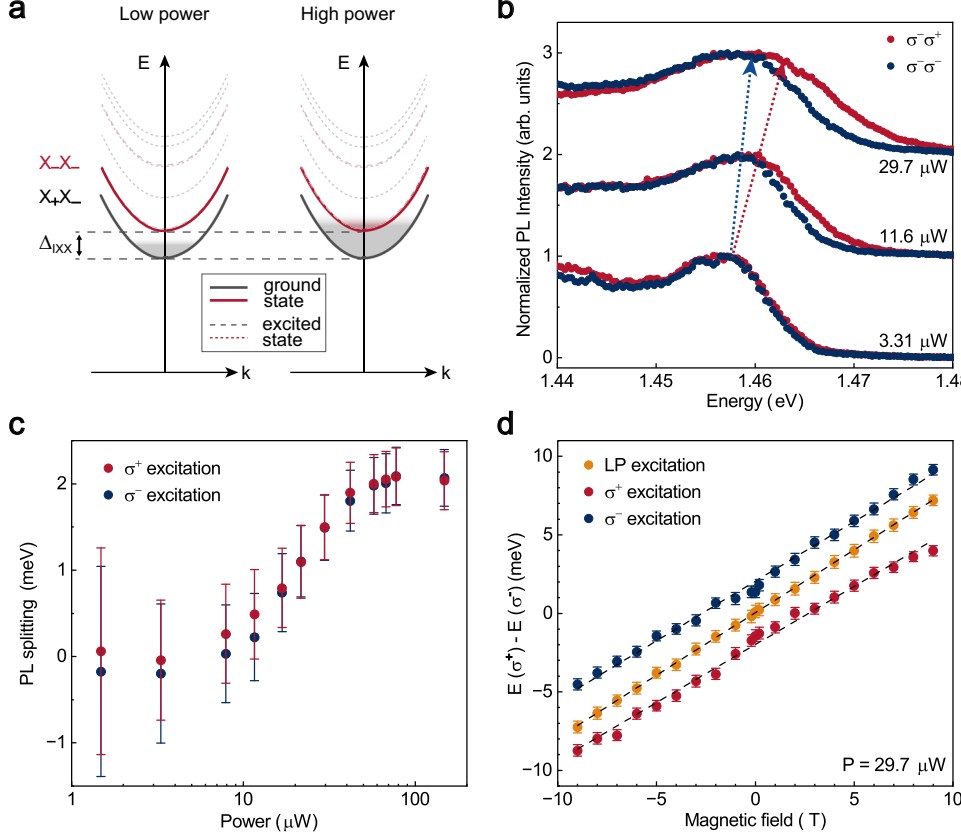

**Fig. 3 | Observation of fine structure of the IXX. a** Energy level diagram of IXX at low and high power. Due to the confinement within the moiré potential, discrete energy levels exist (ground state and excited state). Because of the exchange interaction, the energy of $X_-X_-$ is higher than that of $X_+X_-$, with the energy difference denoted as $\Delta_{IXX}$. The intervalley biexciton can emit the light of $\sigma^+$ and $\sigma^-$. At low power, only the $X_+X_-$ exists, resulting in the same emission peak position of $\sigma^+$ and $\sigma^-$. However, at higher power, the $X_-X_-$ emission also begins to appear, leading to the total $\sigma^+$ light shift to a higher peak position. **b** Normalized interlayer exciton PL spectra of high-energy edge, exciting by $\sigma^-$ light at selected power. **c** PL splitting of IXX as a function of power at circular excitation. Error bars are derived from the fitting uncertainty of the energy of two circularly polarized emissions. **d** Zeeman splitting as a function of magnetic field under circular and linear polarized optical excitation at $P = 29.7\,\mu W$. All error bars are uncertainties in the peak positions extracting from peak fitting.

the energy difference between the cross-polarized and co-polarized PL emission with respect to the excitation helicity. As seen, both cases exhibit the same trend regardless of the excitation helicity. As the excitation power increases, the splitting increases from near zero, reaching a saturation value of 2 meV when the excitation power exceeds 50 μW. Considering that $\sigma^+$ emission also partially comes from $X_+X_-$ (as shown in the Fig. 3a), it follows that the actual fine structure splitting $U_{intra} - U_{inter}$ between the $X_-X_-$ and the $X_+X_-$ is larger than the observed splitting of 2 meV. In addition, comparing with Fig. 2b, one can also note that the increase in peak energy splitting follows a very similar trend to that of the valley polarization of IXX (see Supplementary Section 14). This correlation between valley polarization and peak energy splitting further reinforces the physical picture that the increased valley polarization observed in Fig. 2a is also due to the different IXX occupation within the fine structure.

This fine structure of the IXX can be confirmed through the Zeeman effect as well. Figure 3d plots the Zeeman splitting $\Delta E(IXX) = E(\sigma^+) - E(\sigma^-)$ of the IXX at different out-of-plane magnetic fields, under the excitation with linearly-polarized (LP), $\sigma^+$-polarized and $\sigma^-$-polarized light, respectively. While using $\sigma^-$ light to selectively excite $-K$, the contribution of $X_-X_-$ biexcitons lead to a higher energy for the $\sigma^+$ emission, resulting in an overall upward shift in $\Delta E(IXX)$. Conversely, under $\sigma^+$ excitation, the contribution of $X_+X_+$ lead to a higher energy for the $\sigma^-$ emission, resulting in a downward shift of $\Delta E(IXX)$. In contrast, under LP excitation, the valley remains unpolarized and the corresponding valley splitting at 0 T is zero. By fitting the

Zeeman splitting, we also obtained an energy splitting of approximately 2 meV under circularly-polarized excitation. All above observations confirm the fine structure of the moiré trapped IXX and reveal the high valley polarization being strongly related to the IXX's occupation on this fine structure.

Additionally, as shown in Fig. 3a, the exciton density also influences the occupation of IXX. As demonstrated in Supplementary Section 11, even when IXX is occupied in both levels under high excitation powers (initially high valley polarization), the IXX exciton density decreases with time delay, transitioning to a state where it only occupies the intervalley states $X_+X_-$ (weak valley polarization). Intervalley scattering dominated by the exchange interaction plays a significant role in this process. Given the remained constant value of the ratio $\tau/\tau_v$ of exciton lifetime to valley lifetime (Supplementary Section 11), we can qualitatively exclude the contribution of reduced valley lifetime to the increase in IXX's valley polarization. However, a quantitative discussion of exchange interaction is still necessary.

According to recent work, the exchange interaction plays a significant role in the intervalley scattering of interlayer exciton[33]. To evaluate the strength of the exchange interaction under different excitation powers, we also performed to the magnetic-field-dependent DCP measurements. From the results, it can be determined that the exchange interaction strength of IXX remains constant at about 0.35 meV, whereas that of IX increases from 0.06 meV to 0.22 meV with the increasing excitation power. This result also indicates that the increase in DCP of IXX is not affected by exchange interactions, which

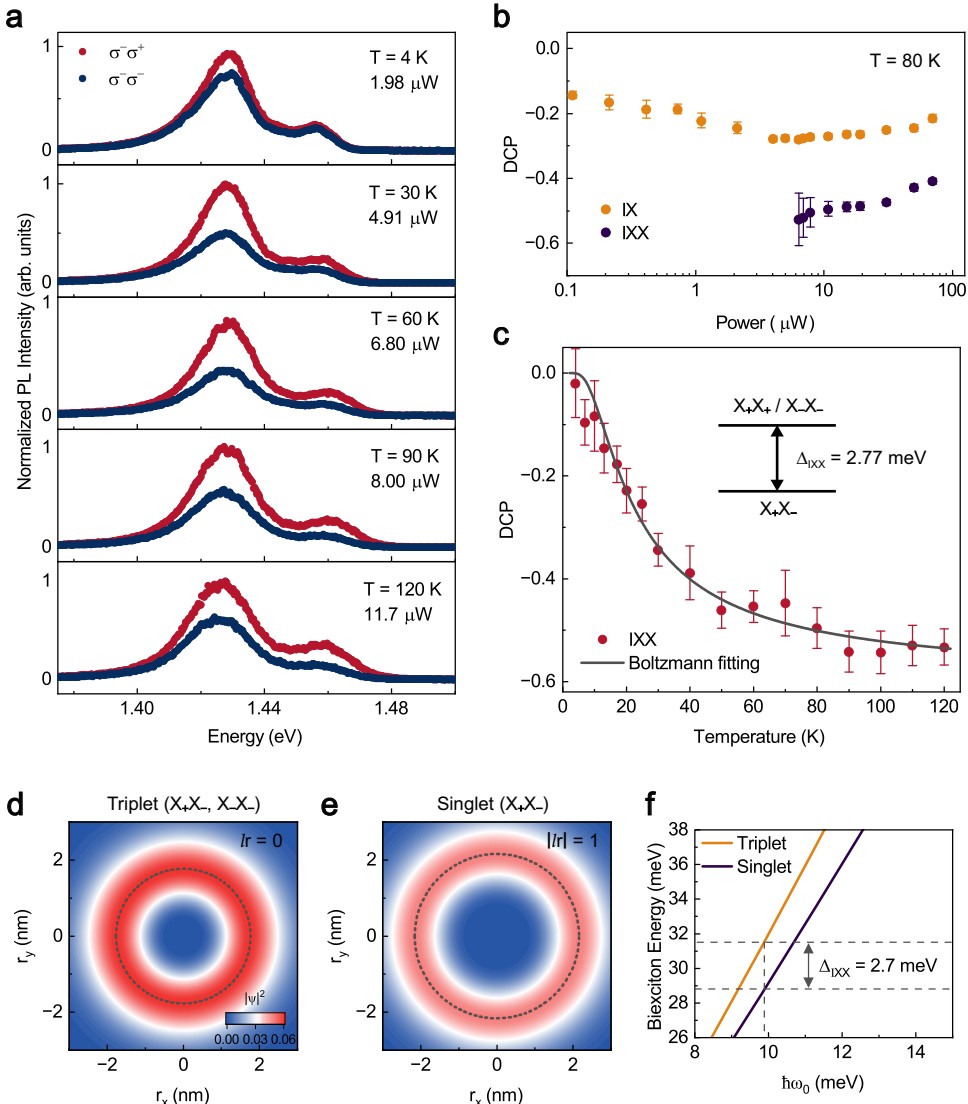

**Fig. 4 | Temperature-dependent degree of polarization. a** The polarization-resolved photoluminescence spectra at different temperatures. **b** The polarization as a function of power at 80 K. Error bars are derived from the fitting uncertainty of the PL intensity of two circularly polarized emissions. **c** Polarization of IXX as a function of temperature, fitted using the Boltzmann equation (solid line). Insets: Schematics of fine structure of IXX. Error bars are derived from the fitting uncertainty of the PL intensity of two circularly polarized emissions. **d**, **e** The probability density of relative coordinates for triplet (**d**) and singlet (**e**) IXX at $\hbar\omega_0 = 9.8$ meV and $M = 10\,m_0$. The singlet state wavefunction ($|l_r| = 1$) exhibits exchange anti-symmetry, thus representing an intervalley biexciton, while the triplet state ($l_r = 0$) exhibits exchange symmetry, representing either an intravalley biexciton or an intervalley biexciton. **f** The biexciton energy of triplet and singlet IXX when $M = 10\,m_0$. The fine structure splitting is 2.7 meV at $\hbar\omega_0 = 9.8$ meV, consistent with experimental measurement.

is consistent with our conclusion. A detailed discussion can be found in the Supplementary Section 15.

**Temperature-dependent valley polarization of the biexcitons**

Building on the previous discussion, we noted that fine structure splitting $U_{\text{intra}} - U_{\text{inter}}$ is a few meV. Consequently, increasing the temperature can easily compensate for this energy difference through thermal activation, thereby tuning the occupation of IXX on the fine structure. Given this intriguing behavior, it is valuable to explore the effect of temperature on the DCP of IXX. Figure 4a shows the helicity-resolved PL spectra as the temperatures varying from 4 K to 120 K, where a $\sigma^-$-polarized light is utilized to selectively excite the $-K$ valley. In order to minimize the effect of high excitation power on the valley polarization, the measurements were carried out at low power regime. For this purpose, the excitation power needs to gradually increase with the temperature to compensate for the loss, as the lifetime of the interlayer excitons decreases with the rising temperature[34].

Additionally, one can observe in Fig. 4a that the emission peak position exhibits only a slight redshift from 4 K to 120 K, potentially attributable to the suppressed electron-phonon coupling at low temperatures. In contrast, a large redshift of approximately 69 meV is observed from 120 K to room temperature (Supplementary Section 16), well consistent with the previous report[35].

Specifically, at 4 K, the $\sigma^+$ and $\sigma^-$ PL emission intensities of IXX are nearly identical, *i.e.*, the DCP of IXX is near zero, corresponding to the net contribution of $X_+X_-$. However, as the temperature gradually increases, the $\sigma^+$ PL emission intensity becomes stronger and stronger than that of $\sigma^-$ emission, showing that the valley polarization of the IXX, namely the contribution of $X_-X_-$, increases with temperature. To better demonstrate the role of temperature, the DCP of IX and IXX at 80 K under different excitation powers are presented in Fig. 4b (The data at 20 K and 50 K are provided in Supplementary Section 17). At 80 K, the negative DCP of IXX varies only slightly with power; however, it can still reach to be about $-50\%$ under the small power of 10 μW.

This result completely differs from Fig. 2b at 4 K, and the high value of DCP of IXX at 80 K strongly indicates that temperature can effectively control the occupation of IXX within the fine structure.

The DCP of IXX as a function of temperature is shown in Fig. 4c. As the temperature rises from 4 K to 120 K, the valley polarization of IXX firstly enhances rapidly from near zero to −50% at around 80 K, and then gradually saturates at a value close to −55%. Qualitatively, we estimate the Hubbard gap with a thermal-activation function[36]. Considering that the IXX's occupation on its fine structure can be described by $e^{\Delta_{\mathrm{IXX}}/k_{\mathrm{B}}T}$ ($\Delta_{\mathrm{IXX}} = U_{\mathrm{intra}} - U_{\mathrm{inter}}$, $k_{\mathrm{B}}$ is Boltzmann constant), the temperature dependent valley polarization of IXX can be described by DCP $= \frac{1}{1+\Gamma e^{\Delta_{\mathrm{IXX}}/k_{\mathrm{B}}T}}$ (Supplementary Section 18), where $\Gamma$ and $\Delta_{\mathrm{IXX}}$ are fitting parameters. While estimating the Hubbard gap using the thermally activated model has considerable uncertainty, we note that we obtain the $\Delta_{\mathrm{IXX}} = 2.77 \pm 0.19$ meV by fitting the experimental data in Fig. 4c with the above expression, which is in close agreement with the peak energy splitting observed in Fig. 3b. This result further confirms that the occupancy of biexcitons in the fine structure can modulate valley polarization.

Additionally, we also measured the magnetic-field-dependent DCP at different temperatures, as shown in Supplementary Section 15. It is found that the exchange interaction strength of IXX remains largely constant with the increasing temperature, unlike the slight increase in the exchange interaction strength of IX. Furthermore, the exchange interaction strength of IXX is always greater than that of IX across all temperatures and excitation powers. This strong exchange interaction of IXX probably leads to the promoted thermal equilibrium within its energy levels.

## The theoretical calculation of the biexciton fine structure

In previous studies, significant overlap of the two exciton wavefunctions resulted in strong Coulomb repulsion[7], leading to different onsite interactions of $U$ for intravalley and intervalley biexcitons. However, the presence of separate energy levels in moiré potential may further reduce the wavefunction overlap, thereby decreasing the energy splitting. Therefore, we calculate intravalley and intervalley biexcitons based on this picture.

We approximate the moiré potential as a harmonic potential $U_0 = \frac{1}{2}M\omega_0^2 r^2$ and treat the Coulomb interaction as a perturbation to calculate the spatial wave functions of biexcitons. Note that the biexciton energy calculation includes the characteristic energy $\hbar\omega_0$, which can be derived from $U_0 \sim \frac{1}{2}M\omega_0^2(\frac{r_{\mathrm{m}}}{2})^2$, where $U_0 = 90$ meV is the moiré potential and $r_{\mathrm{m}} = 7.6$ nm is the moiré periods[21,30]. Here, $M$ represents the exciton effective mass in the heterostructure. It has been pointed out that the band renormalization of the heterostructures yields $M$ increases significantly, leading to our estimate of $M = 10\,m_0$ (Supplementary Section 19). Substituting these values yields the result $\hbar\omega_0 \sim 9.8$ meV.

Before calculating the energy of biexciton numerically, it is necessary to discuss the symmetry of the biexcitons. For intravalley biexcitons, the valley pseudospin $|\uparrow\uparrow\rangle$ requires a symmetric spatial wave function under bosonic exchange. whereas for intervalley biexcitons, the valley pseudospin may be either the antisymmetric singlet $\frac{|\uparrow\downarrow\rangle - |\downarrow\uparrow\rangle}{2}$ or symmetric triplet $\frac{|\uparrow\downarrow\rangle + |\downarrow\uparrow\rangle}{2}$, resulting in antisymmetric spatial wave function for singlet state or symmetric spatial wave function for triplet state, respectively. We designate these as singlet (spatially antisymmetric) and triplet (spatially symmetric) states, which exhibit distinct angular momentum quantum numbers $l_r$: i.e., $l_r = \pm 1$ for singlet state and $l_r = 0$ for triplet state.

We calculated the probability density of relative coordinate $\mathbf{r} = \mathbf{r}_1 - \mathbf{r}_2$ ($\mathbf{r}_1$ and $\mathbf{r}_2$ are the coordinates of two interlayer single excitons IX) of both the triplet and the singlet states, as presented in Fig. 4d and Fig. 4e, respectively. As seen, the triplet state exhibits a smaller average exciton distance and a peak probability density at $r = 1.77$ nm,

while the singlet state shows a notably larger average distance peaking at $r = 2.16$ nm. Consequently, while the singlet state possesses a higher ground-state energy due to its non-zero angular momentum, its diminished Coulomb interaction energy results in the lower total energy relative to that of the triplet state. Detailed calculation can be found in Supplementary Section 19. Figure 4f displays the corresponding energy levels of the triplet and the singlet states. The average biexciton energy is around 30 meV, and the energy difference between the singlet and the triplet states is 2.7 meV, consistent well with our experimental results.

In summary, we studied the unusually high valley polarization and the fine structure of the biexciton in WS$_2$/WSe$_2$ heterostructure. We found that the increase of circularly polarized excitation power significantly enhances the valley polarization of biexciton, fundamentally driven by the change in the occupancy of biexciton within its fine structure. At low powers, the intervalley biexciton dominates, resulting in a near-zero weakly negative valley polarization. In sharp contrast, as the power increases, the occupation of the intravalley biexcitons rises and prevails, leading to dramatically enhanced valley polarization. The helicity-resolved PL emission energy splitting of biexciton provides direct evidence of this fine structure. The result of temperature-dependent valley polarization also provides the clear evidence on the successful tuning of the biexciton occupation within its fine structure, even at small excitation powers. We also calculated the energy-level structure of biexcitons. The calculations show that the average biexciton energy is around 30 meV and the singlet-triplet energy splitting is about 2.7 meV, which agrees well with our experimental results. Our findings certainly help to deepen the understanding of the physics of moiré trapped biexcitons, and thereby can offer new possibilities for their applications in valleytronics, particularly given the fairly high valley polarization of −55% even at an elevated temperature of 120 K.

## Methods
### Sample fabrication
A PDMS-based dry-transfer technique was employed to transfer mechanically exfoliated samples onto SiO$_2$/Si substrates. Monolayer WSe$_2$ was then stacked on top of monolayer WS$_2$, ensuring proper edge alignment. The bilayer is encapsulated in thick h-BN. Following the stacking process, the sample was annealed in a nitrogen atmosphere at 200 °C for 6 h.

### Optical measurements
Photoluminescence measurements were performed by placing the sample into a closed-cycle cryostat (AttoDry 1000). A supercontinuum white-light laser (NKT Photonics SuperK EXTREME, 77 MHz) was used to provide a range of wavelengths, which were focused onto the sample in a 1 μm$^2$ spot. PL emissions were collected using a spectrometer equipped with a liquid-nitrogen-cooled CCD (PyLon eXcelon 100BRX). Polarization-resolved PL measurements were conducted using linear polarizers, half-wave plates and quarter-wave plates. For magnetic field measurements, a variable magnetic field in the range of ±9 T was applied, and spectra were recorded at a temperature of 4 K.

### TRPL measurements
Time-resolved PL (TRPL) was performed with a pulse laser with the repetition rate of 77 MHz for biexcitons and 7.7 MHz for single excitons. Optical signals were collected using a single-photon detector (EXCELITAS SPCM-AQRH-14-FC).

## Data availability
The data supporting this study's findings are available from the corresponding author upon request. Source data are provided with this paper.

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

## Acknowledgements

This work was supported by the National Key Research and Development Program of China (2024YFA1208102, N.P.), the Natural Science Foundation of China (62574192, H.C.) and Anhui Outstanding Young Scientist Fund (2408085J005, H.C.). The authors also acknowledge support from the robotic AI-Scientist platform of Chinese Academy of Sciences. This work was financially supported by the Quantum Science and Technology-National Science and Technology Major Project (QNMP, Grant No. 2021ZD0303303). We thank Prof. Zhigao Sheng, Prof. Bolin Li, and Ms. Ningfang Wang (High Magnetic Field Laboratory, CAS) for their support in the SHG measurements.

## Author contributions

Y.J., Y.S., and N.P. conceived the concept and designed the experiments. Y.J. and H.C. performed the sample fabrication and characterization. Y.J., Y.S., X.W., H.C., N.P., X.C., and Q.T. performed the data and theoretical analysis. Y.J., Y.S., J.Y., Y.Z., P.L., M.W., X.D., Z.W., X.W. and N.P. contributed to discussing, writing, and reviewing the manuscript.

## Competing interests

The authors declare no competing interests.
