## [Peer Review File · Nature Communications]

Tuning Valley Polarization of Moiré Trapped Biexcitons by Fine Structure Occupation in WS₂/WSe₂ heterostructures

Corresponding Author: Professor Hongbing Cai

Version 0:

Reviewer comments:

Reviewer #1

(Remarks to the Author)

The manuscript by authors Y. Jiang et al., presents a study of interlayer excitonic states in a moiré heterostructure of transition metal dichalcogenide (TMD) semiconductors.

Using power-dependent photoluminescence (PL) measurements, the authors report presence of moiré-trapped interlayer biexcitons (IXX) in WS₂/WSe₂ heterostructure, 30 meV above the interlayer exciton (IX). In addition, using polarization-resolved PL measurements, it is shown that the circular polarization of both IX and IXX states evolves as a function of the incident power. Interestingly, both states exhibit significant negative polarization (>25%) at high powers, which is in contradiction to what has been reported so far in these systems. The authors explain the presence of interlayer biexcitons based on the Bose-Hubbard model for moiré-trapped states. Further, they attribute the negative circular polarization to the fine structure of the biexciton. A temperature-dependent PL study is performed to confirm the validity of this model.

The manuscript is well written and organized. The experimental results and arguments of the authors are convincing. I think this manuscript will add to the ongoing discussions of moiré physics in two-dimensional systems. The paper deserves to be published in Nature Communications, once the authors reply to a few minor points raised as follows:

- 1) The samples have been fabricated using mechanical exfoliation. In general, such samples show residual doping at low temperatures. This could affect the formation of multi-particle states like biexcitons. Could the authors include a PL of both the individual layers (WS₂ and WSe₂) as well as the heterostructure near the energy of their intrinsic intralayer excitons? This would help estimate the doping level of the monolayers based on the intensity of the trions. Also, the quality of the samples (dielectric environment inhomogeneity) can be assessed better by probing the broadening of these states.
- 2) In Fig. 1d, a power-dependent PL of IX and IXX states is presented. At 4K, the data is restricted to 1 microW incident power. Why is this the case? Also, as it is observed later in the manuscript that these states show significant negative circular polarization, I believe it would be important to mention the polarization conditions in this measurement.
- 3) Also on a related note, in Fig. 3b, it is shown that the IXX emission gets broader at higher excitation power due to excitation of different states in the fine structure of the biexciton. Could this broadening be contributing to the fitting parameter of power-dependent PL of IXX in Fig. 1d?
- 4) In Fig. 4a, a temperature-dependent PL is presented for the IX and IXX states. In general, for the intralayer excitons in monolayer TMDs, a temperature-dependent red shift in emission energy is observed at higher temperature (M. M. Ugeda et al, Nature Mater 13, 1091-1095, 2014). Why is such a shift not observed for the interlayer states?

Reviewer #2

(Remarks to the Author)

The manuscript titled "Tuning Valley Polarization of Moiré Trapped Biexcitons by Fine Structure Occupation in WS₂/WSe₂ heterostructures" authored by Jiang et al., describes about the formation of the interlayer biexcitons and its valley selectivity along with the fine structures of moiré-trapped biexcitons. They have analyzed the multiparticle states using power and temperature dependent photoluminescence (PL) as well as investigated its valley polarization. The retention of high valley polarization up to 120 K is particularly noteworthy for potential valleytronic and quantum light source applications.

Despite the experimental efforts and findings of moiré-trapped biexcitons along with the theoretical calculations, we believe the manuscript, in its current state, falls short of the standards required for publication in Nature Communications. We recommend rejection in its present form, while strongly encouraging the authors to address the following major points through significant revision.

Please find our comments below:

1. In the fig 1b, it's a bit confusing to understand the valley configuration along with the layer distribution. As the IXX is forming between both the layers, it is a bit ambiguous. To enhance clarity, I suggest explicitly illustrating the formation pathways through different layers, for example, as reported in ref. Physical Review Letters 134, 256402 (2025).
2. The emergence of blue shifted peak has been reported earlier in WS₂/WSe₂ system in recent years in the references Physical Review Letters 134, 256402 (2025), Nature 610, 478–484 (2022), Science 380, 860–864 (2023) etc. However, in the current report there are quite a few differences such as the power law exponent is significantly higher compared to the earlier reports. Second, there is significant increase in the 'α' value for IXX with temperature. Could the authors provide a more detailed discussion or propose possible mechanisms for these discrepancies compared to earlier works?
3. How is the present case being different from the earlier reported correlated Bosonic pairs as reported in Nat. Phys. 19, 1286–1292 (2023) and Science 380, 860–864 (2023) where the higher energy state has been attributed to correlated IXs. Similarly, the authors did not observe any signature of cascaded emission earlier reported in Physical Review Letters, 129(24), p.247401, Nature Communications 15, no. 1 (2024): 1057. Could the authors elaborate on how their system differs, and offer potential explanations for the absence of cascaded emission?
4. What's the typical FWHM of the IX and IXX reported in the manuscript? How the peak fitting has been performed? Again, the confusion arises from the fact that there had been reports of multiple satellite peaks in the literature assigned to either quantum emitters or charged interlayer excitonic states. The authors should clarify if they want to nullify the possibilities of peaks emerging from those energy states.
5. The peak position of IX and IXX remain almost unaltered with increasing power which contrasts with the Nature 610, 478–484 (2022). What are possible reasons to this?
6. Throughout the manuscript, the legends in the polarization dependent spectra are presented only the detection mode. The excitation mode is reported in the figure caption. This renders interpretation of co- and cross-polarization challenging. The authors are advised to revise the nomenclature for improved clarity.
7. In the expression of DCP, how does the exchange interaction play a role? We encourage a more thorough explanation regarding how exchange interactions influence the DCP, particularly in relation to power and temperature variations.
8. What's the origin of spitting for IX in Supplementary section 8?
9. Can the authors also explicitly explain how both intra and intervalley configuration is possible? Due to charge transfer and H type heterobilayer formation only +K spin down (up) conduction band (WS₂) and same spin valence band of WSe₂ should have effective electron and hole population. This should always give rise to singlet biexciton formation and "zero" DCP.
10. The non-radiative decay to from interlayer excitonic to biexciton state is obvious at higher optical power and it should affect the total decay time measured by TRPL. How does the exciton decay time change with optical power? The authors should formulate a rate equation considering all these decay components and thereby fit the power dependent TRPL data with the solution of this rate equation. This will help them to calculate the biexciton population at higher power and other non-radiative decay paths.
11. The authors have shown interlayer excitons and biexcitons in the same harmonic well configuration In Figure 2c. The biexciton transition to excitonic state should be depicted between two different harmonic wells as articulated in ref. 2D Mater. 11 (2024) 025030. Also, what are the different radiative decay pathways to excitonic states (harmonic well states) from intra and intervalley biexcitons?
12. Minor comment: Supplementary section 6 caption is repeated twice while the supplementary section 7 header is absent. Please revise accordingly.

Reviewer #3

(Remarks to the Author)

Version 1:

Reviewer comments:

Reviewer #1

(Remarks to the Author)

The authors have appropriately addressed most of my concerns. In my opinion, the revised manuscript upholds the scope

and requirements of the journal and should be published.

Reviewer #2

(Remarks to the Author)

We would first like to thank the authors for their detailed responses to the most of our concerns or doubts, as well as undertaking additional experimental works. The new data solve almost all our queries effectively and improves the quality of the manuscript.

We only have a few minor comments remaining; nonetheless, the manuscript is almost ready to be accepted in Nature Communications.

1. In response to comment #2, the authors attribute the slight increase in biexcitonic power law with temperature to the exciton delocalization effect from shallow non-radiative potentials. Is this delocalization related to defect sites? Throughout the manuscript, the potentials have primarily been ascribed to moiré potentials, which appear to facilitate radiative recombination. Consequently, the reference to “non-radiative” in Supplementary 6 creates some ambiguity.

2. The authors propose that exciton-exciton annihilation in IXX accounts for the rapid total lifetime observed in TRPL. However, unlike IX, where multiple light cones exist between two parallel bands (IXX and IX band), this phenomenon does not apply to IXX. We suggest that the faster decay of IXX relative to IX results from non-radiative processes, given the increased Bohr radius arising from strong dipolar repulsion between two excitons.

Reviewer #3

(Remarks to the Author)

Dear Editor:

We are pleased to submit our revised manuscript entitled '*Tuning Valley Polarization of Moiré Trapped Biexcitons by Fine Structure Occupation in WS₂/WSe₂ heterostructures*' by Yufei Jiang *et al*, for your further consideration at ***Nature Communications***.

We are grateful to you for your appreciation of this work and giving us opportunity to revise and improve our manuscript. We are also very grateful to the three reviewers for their positive suggestions and constructive feedback, which have greatly helped us to strengthen this work. We have carefully revised the manuscript as well as the supplementary information by point-by-point addressing all the reviewers' comments. Specifically, in response to their valuable comments, we have systematically conducted extensive additional experiments and analyses, which are detailed below:

- *) PL measurements of WS₂ and WSe₂ intralayer exciton emissions in the monolayer and heterostructure regions, combined with spectral shape analyses to rule out the other possible origins of IXX such as charged exciton or quantum emitters.
- *) Discussion on the influence of linewidth broadening on the power-law exponent and analysis of the mechanism for power-law exponent increase with temperature.
- *) Discussion on the temperature-induced redshift of the interlayer exciton emission.
- *) Supplementing more detailed formation pathways of biexcitons in different layers.
- *) Providing the direct evidence for a cascade emission process through the comparison of the lifetime for IX and IXX under the same measurement conditions.
- *) Additional analysis of IXX and IX peak positions versus excitation power, and the discussion on the origin of the blueshift.
- *) Analyses on the effects of power and temperature on the exchange interaction of both IX and IXX via magnetic-field-dependent PL measurements at different excitation powers and temperatures.
- *) PL lifetime measurements of IX and IXX at different excitation powers.
- *) Developing the fitting model for better depicting and understanding the IX/IXX dynamics and power-dependence, with/without incorporating exciton diffusion.

We believe that the comments/concerns of the reviewers have been fully addressed in the revised version, so we sincerely hope that this revised version is ready to be accepted for publication.

Best regards,

Hongbing Cai

Professor

Hefei National Laboratory for Physical Sciences at the Microscale

University of Science and Technology of China

Email: coldice@ustc.edu.cn

In the following document, we provide detailed answers to the reviewers' comments

Reviewer 1(Comments to the Author):

The manuscript by authors Y. Jiang et al., presents a study of interlayer excitonic states in a moiré heterostructure of transition metal dichalcogenide (TMD) semiconductors. Using power-dependent photoluminescence (PL) measurements, the authors report presence of moiré-trapped interlayer biexcitons (IXX) in WS₂/WSe₂ heterostructure, 30 meV above the interlayer exciton (IX). In addition, using polarization-resolved PL measurements, it is shown that the circular polarization of both IX and IXX states evolves as a function of the incident power. Interestingly, both states exhibit significant negative polarization (>25%) at high powers, which is in contradiction to what has been reported so far in these systems. The authors explain the presence of interlayer biexcitons based on the Bose-Hubbard model for moiré-trapped states. Further, they attribute the negative circular polarization to the fine structure of the biexciton. A temperature-dependent PL study is performed to confirm the validity of this model. The manuscript is well written and organized. The experimental results and arguments of the authors are convincing. I think this manuscript will add to the ongoing discussions of moiré physics in two-dimensional systems. The paper deserves to be published in Nature Communications, once the authors reply to a few minor points raised as follows:

Authors Reply:

We sincerely thank the reviewer for the positive recommendation and constructive comments. In response, we measured the intralayer exciton emission, further validating our conclusions. Additional data analyses have helped to provide a more comprehensive understanding of the biexciton physics in the TMD heterostructure.

Remarks and Questions:

- 1. The samples have been fabricated using mechanical exfoliation. In general, such samples show residual doping at low temperatures. This could affect the formation of multi-particle states like biexcitons. Could the authors include a PL of both the individual layers (WS₂ and WSe₂) as well as the heterostructure near the energy of their intrinsic intralayer excitons? This would help estimate the doping level of the monolayers based on the intensity of the trions. Also, the quality of the samples (dielectric environment inhomogeneity) can be assessed better by probing the broadening of these states.*

Authors Reply:

We thank the reviewer for the valuable suggestion. To further investigate the doping level, quality and excitonic emission of the samples, we have performed additional measurements on the intralayer exciton emission from the heterostructure and respective monolayer regions. These results confirm that the investigated heterostructure is almost electrically neutral, which strengthens our data and conclusion.

Mechanically exfoliated TMDs often exhibit low defect concentrations ($10^9 - 10^{10} \text{ cm}^{-2}$) and tend to be charge-neutral (*Nat. Phys.* 19, 1286–1292 (2023); *Nat. Phys.* 20, 34–39 (2024)). To qualitatively determine the doping of the samples, we measured the intralayer exciton photoluminescence (PL) spectrum of the monolayer regions at 4 K as shown in the Fig. R1a-b (we use 680 nm and 532 nm lasers to excite WSe₂ and WS₂ monolayer regions, respectively). As shown in Fig. R1a, a peak at 2.01 eV dominates the monolayer WS₂ intralayer exciton emission, which accords with the neutral exciton energy commonly observed in other reports (*Nature.* 579, 353–358 (2020); *Nat. Commun.* 10, 107 (2019)). For monolayer WSe₂ intralayer exciton emission (Fig. R1b), there are two peaks. The higher-energy peak at 1.73 eV originates from WSe₂ neutral exciton (*Nature.* 587, 214–218 (2020)). The lower-energy one at 1.65 eV is associated with localized exciton (*Nat. Commun.* 6, 8315 (2015)). No obvious charged exciton emission peak can be observed in these spectra. These results indicate that the constituent monolayer TMDs of the heterostructure are primarily composed of the neutral and the localized excitons, implying that the investigated sample is at low doping level.

We also measured the intralayer emission from the heterostructure region as shown in Fig. R1c, the results are well close to those measured in the monolayer regions. The WSe₂ intralayer exciton emission is still dominated by the localized exciton, and the WS₂ intralayer exciton emission shows strong neutral exciton peak at 2.01 eV and another weak peak around 1.92 eV, which may be related to the hybridized exciton (*Nat. Nanotech.* 16, 52–57 (2021)).

Furthermore, the FWHM of the interlayer and the intralayer excitons also indicate the high quality of our samples. Typically, the monolayer WS₂ and WSe₂ intralayer exciton emission exhibits the full width at half maximum (FWHM) ranging from a few meV to about a dozen meV (*Nat. Commun.* 6, 8315 (2015); *Nat. Commun.* 9, 3719 (2018)). Here the measured FWHM for the neutral exciton emissions in monolayer WS₂ and WSe₂ are 16 meV and 13 meV, respectively, well in line with the typical values. Additionally, the FWHM of the interlayer exciton emission measured at low powers is approximately 17 meV, which is also well consistent with the previously reported value ranges from 14 meV to 48 meV (*Nano Lett.* 24, 2773-2781 (2024); *Nat. Phys.* 20, 34–39 (2024)). These results indicate the pretty good quality of our sample.

The discussion here has been included in the main text and **Supplementary Section 2**.

Fig. R1 Intralayer exciton luminescence in the monolayer and heterostructure regions at 4 K. **a** The intralayer exciton photoluminescence spectrum of monolayer WS₂. It is dominated by the neutral exciton emission with an energy of 2.01 eV and a linewidth of 16 meV. The inset shows the power-dependent PL intensity of the neutral exciton emission in WS₂, the fitted power-law exponent is 0.99. **b** The intralayer exciton PL spectrum of monolayer WSe₂. It composes of a neutral exciton emission at 1.73 eV with the linewidth of 13 meV and a localized exciton emission at 1.65 eV with the linewidth of 36 meV. The fitted power-law exponents are 0.95 (1.73 eV) and 0.69 (1.65 eV), respectively, as shown in the inset. **c** The corresponding intralayer exciton emissions in the heterostructure region.

- In Fig. 1d, a power-dependent PL of IX and IXX states is presented. At 4K, the data is restricted to 1 microW incident power. Why is this the case? Also, as it is observed later in the manuscript that these states show significant negative circular polarization, I believe it would be important to mention the polarization conditions in this measurement.*

Authors Reply:

We sincerely thank the reviewer for raising this question regarding to the data range and acquisition conditions, for which the original manuscript was lacking clarity. Prompted by the reviewer's comment, the full spectrum, with the excitation powers spanning from 0.01 μ W to 0.1 mW, is provided in Fig. R2. As illustrated, the interlayer biexciton (IXX) exhibits a superlinear rise at moderate powers. However, at high powers, the filling effect of moiré exciton becomes more significant, leading to the saturation of both interlayer single exciton (IX) and IXX intensities, as reported in prior studies (*Nat. Mater.* 24, 527–534 (2025); *Nat. Phys.* 19, 1286–1292 (2023)). Accordingly, we selected the PL intensities beyond but not much far away from the threshold power for fitting the power dependence. The discussion has been added in **Supplementary Section 6**.

We appreciate the reviewer's suggestion to mention the polarization conditions. To emphasize and clarify the polarization conditions, we have marked the excitation mode in figures (such as " $\sigma^- \sigma^+$ ", which represents excitation with σ^- and collection with σ^+). It should be mentioned that our heterostructure is of the H-type, which differs from the R-type and thus exhibits distinct optical selection rule that is opposite to those in the monolayer TMDs (*Phys. Rev. Lett.* 134, 256402 (2025); *Nat. Mater.* 22, 599-604 (2023)). Once we excite the $-K$ valley of monolayer WSe₂ in the heterostructure with a σ^- circularly polarized light, it leads to the formation of interlayer exciton in the $-K$ valley. Due to the opposite optical selection rule for the interlayer exciton compared to the intralayer ones, the $-K$ valley interlayer exciton emits σ^+ light, which naturally results in the negative polarization. As for the IXX, since its population is predominantly in the intervalley state at low powers, its emission exhibits near-zero polarization. Whereas at high powers, both intervalley and intravalley states coexist, resulting in a much higher negative valley polarization in the IXX emission.

Fig. R2 The power dependence of IXX and IX intensities over a wide excitation power range at 4 K. At low powers, IXX retains its characteristic of superlinear rise, exhibiting a power-law exponent of 1.40. At high powers, both IX and IXX exhibit saturation.

3. *Also on a related note, in Fig. 3b, it is shown that the IXX emission gets broader at higher excitation power due to excitation of different states in the fine structure of the biexciton. Could this broadening be contributing to the fitting parameter of power-dependent PL of IXX in Fig. 1d?*

Authors Reply:

We thank the reviewer for raising this point regarding to the spectral linewidth broadening of the IX and IXX emissions. We agree that this is an important point to clarify. We think that the linewidth broadening does not significantly impact the extracted power-law exponents in our analysis.

As shown in Fig. 1d and noted in the R1Q2 (the second question raised by the

reviewer 1), our power-law fitting was performed specifically near the threshold power (from 0.2 μW to 1 μW). At this range, both a previous report (*Nat. Phys.* 20, 34-39 (2024)) and our double-peak fitting in Fig. R3 demonstrate that the FWHM reaches a minimum and remains relatively stable. Therefore, the FWHM does not change significantly at this stage, the extracted power-law exponent is not substantially influenced. Additionally, while the IXX linewidth has a larger error at low powers due to its weak intensity, using the peak height near the threshold provides the most reliable measure to capture its characteristic superlinear behavior (*Nat. Commun.* 14, 6910 (2023)).

Moreover, even if we consider the increased broadening of the IXX emission, such an effect would tend to increase the extracted power-law exponent by only about 0.2, which does not affect our conclusion that power-law exponent of the biexciton is close to 2.

The discussion here has been included in **Supplementary Section 6**.

Fig. R3 The power dependence FWHM of the IXX and IX emissions. For IX and IXX, the broadening changes primarily occur at high powers, while the broadening changes are minimal near the threshold power. The dashed line represents the minimum FWHM of interlayer single exciton.

4. *In Fig. 4a, a temperature-dependent PL is presented for the IX and IXX states. In general, for the intralayer excitons in monolayer TMDs, a temperature-dependent red shift in emission energy is observed at higher temperature (M. M. Ugeda et al, Nature Mater 13, 1091-1095, 2014). Why is such a shift not observed for the interlayer states?*

Authors Reply:

We thank the reviewer for this valuable feedback regarding to the little spectral redshift. The absence of a pronounced redshift in Fig. 4a can be attributed to the low-temperature conditions (4 K to 120 K) in our experiment. In fact, previous studies have shown that at low temperatures, the redshift of the excitonic emission peak with the rising temperatures is much smaller than that at high temperatures due to the

suppression of electron-phonon coupling (*Phys. Rev. B.* 105, 085111 (2022)). In these reports, the energy redshift within the 4 K to 120 K temperature range is on the order of a few meV to a dozen meV (*J. Phys. D Appl. Phys.* 49, 465101 (2016); *Npj 2D Mater. Appl.* 2, 30 (2018); *Nano Lett.* 24, 8671-8678 (2024)). In alignment with these works, we fitted the spectra at 4 K and 120 K and obtained a redshift of 2 meV in our experiment (Fig. R4a).

For comparison with a previous report (*Nat. Mater.* 13, 1091-1095 (2014)), we also measured the interlayer exciton PL at room temperature (Fig. R4b). The peak position exhibited a blueshift of ~ 69 meV compared to that at 120 K, consistent with the ~ 80 meV shift observed between 77 K and room temperature in the literature.

The discussion here has been included in **the main text and Supplementary Section 16.**

Fig. R4 Comparison of peak positions at different temperatures. **a** The photoluminescence of interlayer exciton at 4 K and 120 K. The IX peak position is 1.429 meV at 4 K and 1.427 meV at 120 K, indicating a redshift of 2 meV. Black and red dashed lines mark the peak energy of 4 K and 120 K. **b** Comparison of photoluminescence at room temperature and 120 K. The deviation from a single-peak emission at the high-energy side originates from the broadening of intralayer excitons. The emission peak position at room temperature is 1.358 meV, which is redshifted by approximately 69 meV relative to the peak position at 120 K.

We would like to express our gratitude once again for the reviewer's insightful comments, which have greatly helped us to improve the quality of this work. In response to the reviewer's comments, all the points have been carefully addressed, and the corresponding revisions have been made in the revised manuscript, including the intralayer exciton PL spectra in the monolayer and heterostructure regions (Fig. R1), power-dependent PL intensities over a wide excitation power range (Fig. R2), the power dependence FWHM of IXX and IX emissions (Fig. R3) and the peak positions at different temperatures (Fig. R4). We believe these substantial improvements provide more clarity and more comprehensive understanding of this work, which we hope to meet the reviewer's expectations.

Reviewer 2 (Comments to the Author):

The manuscript titled “Tuning Valley Polarization of moiré Trapped Biexcitons by Fine Structure Occupation in WS₂/WSe₂ heterostructures” authored by Jiang et al., describes about the formation of the interlayer biexcitons and its valley selectivity along with the fine structures of moiré-trapped biexcitons. They have analyzed the multiparticle states using power and temperature dependent photoluminescence (PL) as well as investigated its valley polarization. The retention of high valley polarization up to 120 K is particularly noteworthy for potential valleytronic and quantum light source applications.

Despite the experimental efforts and findings of moiré-trapped biexcitons along with the theoretical calculations, we believe the manuscript, in its current state, falls short of the standards required for publication in Nature Communications. We recommend rejection in its present form, while strongly encouraging the authors to address the following major points through significant revision.

Authors Reply:

We greatly appreciate the reviewers for considering our work as “particularly noteworthy for potential valleytronic and quantum light source applications”. We also thank the reviewers' thoughtful and valuable feedback, which has greatly helped us to clarify and strengthen the discussion of biexciton in our manuscript. In response to these insightful comments, we conducted additional experiments and discussion, which provide further evidence to support our conclusions and address the raised concerns. We deeply appreciate the reviewer's valuable contributions, which have significantly enhanced the quality and impact of this study.

Remarks and Questions:

- 1. In the fig 1b, it's a bit confusing to understand the valley configuration along with the layer distribution. As the IXX is forming between both the layers, it is a bit ambiguous. To enhance clarity, I suggest explicitly illustrating the formation pathways through different layers, for example, as reported in ref. Physical Review Letters 134, 256402 (2025).?*

Authors Reply:

We thank the reviewers for this very constructive suggestion. To facilitate a better understanding of the biexciton's valley property and formation pathway, we have added a more detailed schematic illustration of the biexciton formation along with the valley configuration and the layer distribution in Fig. R5. This new schematic provides a visual foundation for our discussion, which can greatly enhance the clarity.

Due to the Type-II band alignment in WS_2/WSe_2 heterostructure, as long as an intralayer exciton is excited in the $-K$ valley of WSe_2 , the electron undergoes a rapid charge transfer process to the $+K$ valley of WS_2 , forming an interlayer exciton with its electron in the WS_2 layer and its hole in the WSe_2 layer (we denote an interlayer exciton IX with a hole in the $-K$ ($+K$) valley of WSe_2 and an electron in the $+K$ ($-K$) valley of WS_2 as a $-K$ ($+K$) valley interlayer exciton), as shown in Fig. R5a. Due to intervalley scattering, IX can remain in the $-K$ valley (Fig. R5a) or scatter into the $+K$ valley (Fig. R5b), leading to a distribution in both valleys even if only the $-K$ valley of WSe_2 is initially excited. This explains the observed weak valley polarization of the IX.

When another exciton is excited into the same moiré potential (Fig. R5c and Fig. R5d), the two IXs can combine together to form an interlayer biexciton (IXX). Depending on the valley type of the constituent interlayer excitons, there are two types of interlayer biexcitons: 1. Under selective excitation of the $-K$ valley, the $-K$ valley exciton may combine with a $-K$ exciton without intervalley scattering, as shown in Fig. R5c. Here, the moiré potential contains two combined interlayer excitons both in the $-K$ valley. This forms an intravalley biexciton. 2. Alternatively, the $-K$ valley exciton can also combine with an exciton of $+K$ valley that undergo intervalley scattering, as shown in Fig. R5d. In this case, the moiré potential hosts two interlayer excitons occupying the $+K$ and $-K$ valleys, respectively. This forms an intervalley biexciton.

To more intuitively illustrate the layer distribution and the distinct selection rules of the IXX, we have updated the Figure 1b, and the discussion here has been included in **Supplementary Section 8**.

Fig. R5 Interlayer exciton formation processes related to layers. **a** Formation of interlayer single excitons. When the $-K$ valley of WSe_2 is excited by σ^- circularly polarized light (yellow solid arrows), $-K$ valley interlayer excitons are formed due to charge transfer (yellow dashed arrows), which emit σ^+ circularly polarized light. The black arrows indicate the spins of electrons and holes. **b** $-K$ valley excitons undergo intervalley scattering into the $+K$ valley, which emit σ^- circularly polarized light. **c** Formation of intravalley biexcitons. Under selective excitation of the $-K$ valley, the $-K$ valley exciton can also combine with an exciton that remains in the $-K$ valley without scattering (**a**). Here, the moiré potential contains two combined interlayer excitons both in the $-K$ valley. Therefore, intravalley biexciton is polarized emission. **d** Formation of intervalley biexcitons. Under selective excitation of the $-K$ valley, the $-K$ valley exciton may combine with a $-K$ exciton that has undergone intervalley scattering (**b**). Here, the moiré potential hosts two interlayer excitons occupying the $+K$ and $-K$ valleys, respectively. Therefore, intervalley biexciton is unpolarized emission.

2. *The emergence of blue shifted peak has been reported earlier in WS2/WSe2 system in recent years in the references Physical Review Letters 134, 256402 (2025), Nature 610, 478–484 (2022), Science 380, 860–864 (2023) etc. However, in the current report there are quite a few differences such as the power law exponent is significantly higher compared to the earlier reports. Second, there is significant increase in the ‘ α ’ value for IXX with temperature. Could the authors provide a more detailed discussion or propose possible mechanisms for these discrepancies compared to earlier works?*

Authors Reply:

We appreciate the reviewers' concern regarding the power-law exponents. In response, we have further investigated the observation that the power-law exponent increases with temperature and have thoroughly revised the manuscript accordingly. This discussion greatly enriches our study.

Firstly, in this study, the power-law exponent of the IXX measured at 4 K is 1.40. This value is unambiguously greater than 1 for the interlayer exciton and is well consistent with the reported biexciton's power-law exponent ranging from 1.3 to 2 (*e.g.*, *Phys. Rev. Lett.* 134, 256402 (2025); *Nat. Mater.* 19, 624-629 (2020)).

Secondly, we observed a significant increase in the power-law exponent with rising temperature. We attribute this behavior to the exciton delocalization effect from shallow non-radiative potential. At low temperatures, a considerable portion of IX and IXX are trapped by the possibly existing non-radiative shallow potential wells, which hardly contributes to the luminescence and simultaneously suppresses the overall formation of the IX and the IXX, thereby reducing their power-law exponents. As the temperature increases, these trapped IX and IXX are delocalized and more contributive to the luminescence, therefore leading to the increase toward the ideal values of 1 and 2 for the IX and the IXX, respectively (*Nat. Commun.* 14, 6910 (2023)).

Moreover, this delocalization mechanism may also explain why the IXX' power-law exponent exceeds 2 at some temperatures. At high temperatures, when excitons are delocalized from shallow potential wells, it may become an additional generation channel for the IXX, thereby enhancing its emission beyond the conventional quadratic dependence.

The discussion here has been included in **Supplementary Section 6 and main text**.

- 3. How is the present case being different from the earlier reported correlated Bosonic pairs as reported in Nat. Phys. 19, 1286–1292 (2023) and Science 380, 860–864 (2023) where the higher energy state has been attributed to correlated IXs. Similarly, the authors did not observe any signature of cascaded emission earlier reported in Physical Review Letters, 129(24), p.247401, Nature Communications 15, no. 1 (2024): 1057. Could the authors elaborate on how their system differs, and offer potential explanations for the absence of cascaded emission?*

Authors Reply:

We thank the reviewers for this valuable comment. To some extent, here the studied biexciton is consistent with the high-lying states reported in those references

(*Nat. Phys.* 19, 1286–1292 (2023); *Science*. 380, 860–864 (2023)). Moreover, in direct response to the reviewer's suggestion, we have performed additional measurements based on the method in *Phys. Rev. Lett.* 129, 247401 (2022), and have indeed observed the cascaded emission process. Consequently, the scope of this study has been expanded to include these new results, providing a more comprehensive picture. A point-by-point discussion is detailed as follows.

Firstly, several recent studies (*Nat. Phys.* 19, 1286–1292 (2023); *Science* 380, 860–864 (2023)) have reported that when the density of interlayer excitons matches the moiré density, the interlayer excitons periodically occupy the moiré superlattices, leading to the formation of correlated bosonic states. At the exciton density exceeding the moiré density, two interlayer excitons can simultaneously occupy within one moiré potential, which gives rise to the high-energy doubly-occupied states (named as the biexcitons in this study). On one hand, similar to these studies, we also observed the high-energy doubly-occupied states as the exciton density exceeds the moiré density. On the other hand, distinct from these prior studies which primarily explore the formation and periodic occupation of correlated states at lower densities, our focus here is to probe the more and unique behaviors of biexcitons in the higher density regime, such as high degree of polarization and its control.

In addition, we also observed the indications of correlated states. For example, the spectral linewidth of the IX is minimized exactly near the exciton density corresponding to the moiré density (Fig. R6), which is well consistent with strong correlation phenomena reported in other literatures (*Nat. Phys.* 20, 34-39 (2024)).

Fig. R6 The power-dependent FWHM of the IX. When the exciton density equals the moiré density, exciton correlation states may form, suppressing exciton diffusion and causing the FWHM to approach its minimum (indicated by the black dashed line).

Secondly, we also observed the cascade emission of the IXX. In previous works (*Phys. Rev. Lett.* 129, 247401 (2022); *Nat. Commun.* 15, 1057 (2024)), the cascade emission processes are attributed to the relaxation between discrete IX energy levels formed by moiré confinement. However, unlike these former cases, the cascade emission process of IXX involves relaxation between IXX, IX, and the ground state. The Fig. R7a and Fig. R7c illustrates the three-level diagram of the ground state, interlayer single exciton, and interlayer biexciton, where the high-energy biexciton state will relaxes to the IX state after one constituent interlayer exciton recombines.

To verify this process, we performed additional time-resolved photoluminescence (TRPL) under different excitation powers and observed the IXX cascade emission. At the low powers where the biexciton is absent, the IX dynamics reflects only its intrinsic relaxation (Fig. R7b). In sharp contrast, at the high powers where both the IX and the IXX exist, while IXX undergoes its own relaxation intrinsically (Fig. R7d), its decay always concomitantly leads to an increase in the IX population, manifesting as a rising component in the IX lifetime trace. More importantly, the decay time of IXX (to $1/e$ of its maximum) perfectly matches the rise time of IX to its maximum value—a clear signature of the biexciton cascade emission. Thanks to the valuable suggestion, these discussions have significantly strengthened our manuscript, which have been included in **Supplementary Section 7 and main text**.

Fig. R7 Biexciton cascade emission. **a** Three-level energy diagram of biexciton, exciton, and ground state at low powers, where biexcitons are absent. **b** Time-resolved photoluminescence at low powers. the upper panel shows the time-resolved photoluminescence of IX, exhibiting only a decay process, while the lower panel shows the time-resolved photoluminescence of IXX (no emission). The yellow area represents the instrument response function. **c** Three-level energy diagram at high powers. **d** Time-resolved photoluminescence at high powers. the lower panel shows the time-resolved photoluminescence of IXX, and the upper panel shows the time-resolved photoluminescence of IX, including both injection and decay processes. The yellow area represents the instrument response function. The time for IXX relaxation to $1/e$ is the same as the time for IX to rise to its maximum value (as indicated by the dashed line).

4. *What's the typical FWHM of the IX and IXX reported in the manuscript? How the*

peak fitting has been performed? Again, the confusion arises from the fact that there had been reports of multiple satellite peaks in the literature assigned to either quantum emitters or charged interlayer excitonic states. The authors should clarify if they want to nullify the possibilities of peaks emerging from those energy states?

Authors Reply:

We sincerely thank the reviewers for the valuable suggestion. As shown in Fig. R8a and Fig. R8b, we fitted the interlayer exciton doublet emission with a double-Voigt function and obtained the full width at half maximum (FWHM) ranging from 15 – 30 meV for the IX, while the FWHM approximately 5 – 20 meV for the IXX. This spectral shape and fitted FWHM are consistent with the literatures (*Nat. Phys.* 20, 34-39 (2024)). We observed that the FWHM of the IX gradually decreases with the increasing power until the exciton density reaches the moiré density, and reaching its minimum around 0.2 μ W. This observation aligns well with the finding that the FWHM is minimized when the exciton density equals to the moiré density (*Nat. Phys.* 20, 34-39 (2024)), where the resulting strongly correlated exciton state suppresses exciton diffusion, thereby reducing the broadening of the exciton emission spectrum.

As mentioned by the reviewers, previous studies have reported multiple narrow-linewidth interlayer exciton emission peaks—with the linewidths on the order of μ eV—under the low-power excitation in the nW range (*Sci Adv.* 6, 8526-8537 (2020)), suggesting their potential as quantum emitters. The narrow peaks merge into a broader emission peak at the higher powers (*Nature Reviews Materials.* 7, 778-795 (2022)). In this study, even the interlayer exciton emission under the low-power (a few nW) excitation exhibits a broad peak with a linewidth of approximately 15 meV centered near 1.42 eV; moreover, we did not observe any characteristic peak merging behavior at high powers, thereby excluding the possibility of the quantum emission. In addition, the emission characteristics in this study—including the peak energy and spectral shape—are well in line with the recently reported features for high-quality WS₂/WSe₂ heterostructures (*Phys. Rev. Lett.* 134, 256402 (2025)). Therefore, we think that the IX and the IXX do not originate from quantum emitters.

Besides, mechanically exfoliated TMDs often exhibit low defect concentrations ($10^9 - 10^{10} \text{ cm}^{-2}$) and tend to be charge-neutral (*Nat. Phys.* 19, 1286–1292 (2023); *Nat. Phys.* 20, 34–39 (2024)). For this purpose, we have further measured the intralayer exciton PL of the monolayer WS₂ (Fig. R8c), WSe₂ (Fig. R8d) and heterostructure (Fig. R8e) regions to qualitatively confirm the charge neutrality of the studied TMDs sample. Intralayer exciton PL spectrum of monolayer region is dominated by the neutral exciton with the absence of observable charged exciton emission. Furthermore, the intralayer exciton emission from the heterostructure region is similar to that of the monolayer region (Fig. R8e), with no obvious charged exciton emission is observed. This suggests that the monolayer and the heterostructure regions in our sample are almost electrically

neutral. Therefore, we ruled out the possibility of the IXX emission being charged exciton luminescence. Furthermore, the power-law exponent of 2 observed for IXX suggests that IXX represents biexcitons.

The discussion here enriches our research and has been included in **Supplementary Section 5**.

Fig. R8 The FWHM of interlayer exciton and the emission of intralayer exciton in monolayer and heterostructure regions. **a** Fitting results of interlayer excitons using double Voigt fitting (yellow and blue curves). **b** The FWHM of IX and IXX obtained from the fitting at different powers. The linewidth of IX ranges from 15 – 30 meV, while the linewidth of IXX is approximately 5 – 10 meV at low powers, subject to fitting uncertainties. At high powers, the linewidth of IXX is around 20 meV. **c**, **d** and **e** Intralayer exciton PL spectra in monolayer WS_2 (**c**), WSe_2 (**d**) and heterostructure (**e**) regions. The photoluminescence of the monolayer WS_2 is dominated by neutral excitons (X, 2.01 eV), the PL of the monolayer WSe_2 is dominated by neutral excitons (X, 1.73 eV) and localized excitons (LX, 1.65 eV). The PL from the heterostructure region is similar to that of the individual monolayers. The absence of distinct intralayer charged exciton emission indicates that the sample is almost electrically neutral.

5. *The peak position of IX and IXX remain almost unaltered with increasing power which contrasts with the Nature 610, 478–484 (2022). What are possible reasons to this?*

Authors Reply:

We thank the reviewers for their very careful observations. The blueshift of the peak positions was not sufficiently highlighted in our original manuscript. In response, we performed additional analysis on the peak positions of IX and IXX, as shown in Fig. R9, and we also observed the blueshift of the peak positions. This addition not only addresses the concern but also enriches our systematic analysis of the power-dependent emission behavior.

The observed blueshift is approximately 10 meV, which is notably smaller than the ~40 meV blueshift reported in *Nature* 610, 478–484 (2022). This discrepancy can be attributed to the different configurations employed for the heterostructure. In the referenced study, the heterostructure was suspended over etched holes in a silicon wafer, whereas our sample is fully encapsulated by hexagonal boron nitride (hBN). The higher dielectric constant of hBN enhances screening of the dipole-dipole Coulomb interaction, therefore suppressing the magnitude of the blueshift. This interpretation is also consistent with other literatures on hBN-encapsulated heterostructures, where a blueshift of around 10 meV has often been observed (*Nat. Phys.* 19, 1286-1292 (2023); *Nat. Mater.* 24, 527-534 (2025)), well in line with our results.

Furthermore, the blueshift behavior of interlayer biexcitons in this study differs from that reported in *Nature* 610, 478–484 (2022). In this study, the moiré trapped IXX experiences strong Coulomb repulsion, featuring an energy ~30 meV higher than the energy of the IX. This results in a consistent blueshift trend for the IXX with that of the IX, as shown in Fig. R9. In contrast, in the prior study, the sample was dominated by the emission from lower-energy free biexcitons, even leading to a redshift in the biexciton emission peak. This fundamental distinction in the dominant biexciton species accounts for the key difference between this work and the cited report, which also highlights the importance of moiré potential on manipulating the biexciton's properties in this study.

The discussion here has been included in **Supplementary Section 3 and main text**.

Fig. R9 The emission peak positions the IX and the IXX respectively as a function of the excitation power. As the power increases, the IX and IXX emission peaks blueshift, maintaining an energy difference of approximately 30 meV.

6. *Throughout the manuscript, the legends in the polarization dependent spectra are presented only the detection mode. The excitation mode is reported in the figure caption. This renders interpretation of co- and cross-polarization challenging. The authors are advised to revise the nomenclature for improved clarity.*

Authors Reply:

We thank the reviewers for this considerate suggestion. we have carefully checked the whole manuscript and marked the excitation mode in the related figures (such as “ $\sigma^- \sigma^+$ ”, it represents excitation with σ^- and collection with σ^+ .) to clarify the excitation-emission configurations of cross-polarized and co-polarized emission spectra. This revision enhances the clarity of the manuscript.

7. *In the expression of DCP, how does the exchange interaction play a role? We encourage a more thorough explanation regarding how exchange interactions influence the DCP, particularly in relation to power and temperature variations.*

Authors Reply:

We thank the reviewers for the very constructive suggestion. Recognizing that exchange interaction critically influences the valley properties in TMDs, in response, we have performed additional magnetic-field-dependent DCP measurements under a series of excitation powers and temperatures. These experiments were designed to directly quantify the exchange interaction for both IX and IXX, thereby strengthening

the rigor and depth of our study.

It is well established that in TMDs, the valley depolarization process was dominated by exchange interaction, which shortens the valley lifetime and consequently degrades the valley polarization. The strength of the exchange interaction depends primarily on the electron–hole wavefunction overlap. As demonstrated in *Phys. Rev. Lett.* 134, 256402 (2025), interlayer excitons exhibit significantly weaker exchange interaction compared to intralayer counterparts due to the spatial separation of electrons and holes. Moreover, the exchange interaction is also influenced by the center-of-mass momentum of excitons, as reported in *Phys. Rev. B.* 89, 205303 (2014) and *Nano Lett.* 25, 6708-6715 (2025). Therefore, both temperature and exciton density can effectively modulate the exchange interaction by altering the momentum distribution.

Followed by the reviewers' suggestions, to quantitatively investigate these effects, we carried out magnetic-field-dependent DCP measurements at different powers and temperatures. The representative results acquired at 4 K and 10 μ W are summarized in Fig. R10. Fig. R11 and Fig. R12 summarize the results of power- and temperature-dependent exchange interaction.

Fig. R10 shows the σ^- and σ^+ circularly-polarized PL spectra under σ^- and σ^+ excitation, respectively (Fig. R10a and Fig. R10b). Through fitting the PL intensities of IX and IXX, we can calculate their *DCP* using the formula:

$$DCP(\sigma^- \text{ exc}) = \frac{I(\sigma^-) - I(\sigma^+)}{I(\sigma^-) + I(\sigma^+)} \quad (1)$$

$$DCP(\sigma^+ \text{ exc}) = \frac{I(\sigma^+) - I(\sigma^-)}{I(\sigma^+) + I(\sigma^-)} \quad (2)$$

Where $I(\sigma^+)$ and $I(\sigma^-)$ denote the fitted σ^+ and σ^- polarized PL intensity, respectively. As shown in Fig. R10c and Fig. R10d, both the IX and the IXX exhibit weak valley polarization at zero magnetic field. Under an out-of-plane magnetic field, their DCP rises rapidly and saturates at about ± 2 T, well consistent with the behavior reported in *Phys. Rev. Lett.* 134, 256402 (2025).

Fig. R10 Magnetic field dependent of DCP for IX and IXX at 4 K. **a** and **b** Circularly polarized PL spectra under σ^- (**a**) and σ^+ (**b**) excitation at different magnetic fields, with an excitation power of 10 μ W. **c** and **d** DCP of IX (**c**) and IXX (**d**) under different magnetic fields.

To further analyze the valley polarization, we applied the method described in *Nat. Commun.* 9, 753 (2018) and *Nano Lett.* 25, 6708-6715 (2025), defining the valley polarization (VP) as $VP = [DCP(\sigma^- exc) + DCP(\sigma^+ exc)]/2$. The resulting VP as a function of the magnetic field under different excitation powers is shown in Fig. R11a and Fig. R11b. We fitted these data using the following model (*Nano Lett.* 25, 6708-6715 (2025)):

$$VP = \frac{P_0}{1 + 2 \frac{\tau}{\tau_{v0}} / [1 + (\frac{B}{B_c})^2]} \quad (3)$$

Here τ_{v0} represents the valley lifetime at zero field, τ is the exciton lifetime. B_c is the characteristic magnetic field that reflects the strength of the exchange interaction.

As established in *Phys. Rev. Lett.* 134, 256402 (2025), B_c is directly related to the exchange interaction strength via $J_{ex} \sim g\mu_B B_c$. We extracted B_c for both IX and IXX across different powers (Fig. R11c) and then derived the corresponding J_{ex} values (Fig. R11d). For the IX, J_{ex} increases from 0.06 meV to 0.22 meV with the increasing powers. This trend can be attributed to the enhancement of the center-of-mass momentum under the higher exciton densities, which in turn leads to an enhancement of the exchange interaction. Specifically, the exchange interaction strength in this work is also close to the reported 0.24 meV for the H-type heterostructures in *Phys. Rev. Lett.* 134, 256402 (2025). In contrast, the exchange interaction of IXX remains almost unchanged over the same power range, indicating that the effect does not govern the

DCP variation of the biexciton.

Fig. R11 Exchange interaction measurement at different excitation power. **a** and **b** Magnetic field dependence of valley polarization of the IX (**a**) and the IXX (**b**) at different excitation powers. Hollow circles represent data points, while the solid curve represents the result of fitting the data using the formula. **c** The characteristic magnetic fields B_c of the IX and the IXX at different powers. **d** Exchange interaction strength of the IX and IXX at different powers. As the excitation power increases, the exchange interaction strength of IX increases, while the exchange interaction of IXX remains essentially

Then we performed the similar analysis on temperature-dependent measurements at a fixed power of 16 μW . The temperature-dependent valley polarization of IX and IXX is shown in Fig. R12a and Fig. R12b, and the extracted characteristic magnetic field B_c and exchange interaction J_{ex} are summarized in Fig. R12c and Fig. R12d at different temperatures, respectively. As the temperature increases, the exchange interaction strength of IX increases slightly, which is also consistent with the temperature induced increase of the center-of-mass momentum of the IX. For the IXX, the large uncertainty in the IXX interaction strength makes it difficult to determine whether the trend decreases or remains constant with rising temperature. However, the increase in temperature maybe lead to an increase in the center-of-mass momentum, thus ruling out the possibility of a reduced exchange interaction strength. Therefore, we conclude that the IXX exchange interaction does not change with temperature.

Fig. R12 Exchange interaction measurement at different temperatures and a power of 16 μW . **a** and **b** Magnetic field dependence of valley polarization of IX (**a**) and IXX (**b**) at different temperatures. Hollow circles represent data points, while the solid curve represents the result of fitting the data using the formula. **c** The characteristic magnetic fields B_c of IX and IXX at different temperatures. **d** Exchange interaction strength of IX and IXX at different temperatures. As temperature increases, the exchange interaction strength of IX increases, while the exchange interaction of IXX remains essentially unchanged.

Furthermore, due to strong Coulomb repulsion, the two constituent excitons of the IXX naturally have the larger center-of-mass momentum, hence resulting in a stronger exchange interaction for the IXX than that for the IX. This enhanced exchange interaction may promote the thermal equilibrium population of IXX between its two energy levels.

In summary, we systematically measured the power and temperature dependence of the IX and the IXX as well as the influences on their exchange interactions. Our results demonstrate that the exchange interaction of the IXX remains unaffected by variations in temperature and excitation power. The discussion here has been included in **Supplementary Section 15 and main text**.

8. *What's the origin of spitting for IX in Supplementary section 8?*

Authors Reply:

We thank the reviewers for raising this insightful question regarding to the valley splitting for the IX. In the Fig. R13, we demonstrate the power dependence of valley splitting of the IX. We observed that the valley splitting exhibits two distinct regimes: the stages increasing and decreasing with the increasing power. We will now discuss the potential origins of these two regimes.

In the first stage under the low excitation powers, the valley splitting of the IX increases with the rising excitation powers. This behavior can be attributed to valley-dependent exchange interaction. Previous studies (*Nat. Nanotechnol.* 16, 148-152 (2021)) have reported that when the exciton density of the +K valley (n_{+K}) exceeds that of the -K valley (n_{-K}), the exchange interaction raises the energy of the +K valley above that of the -K valley, thereby inducing the valley splitting. Moreover, the magnitude of this splitting increases with the population difference $\Delta n = n_{+K} - n_{-K}$, due to the strengthening exchange interaction. This mechanism aligns well with our observations: in this regime, the increase in the valley polarization of IX leads to a growing population difference, which in turn drives the enhancement of the valley splitting.

The difference emerges in the second stage, where the IX energy splitting begins to decrease under the higher excitation powers. While the direct evidence accounting for this decline is currently lacking, we speculate that it may be attributed to the Coulomb screening. Theoretical study suggests that the Coulomb screening could play a significant role in exciton-exciton interaction (*Phys. Rev. X.* 14, 031025 (2024)). As the exciton density increases, the enhanced Coulomb screening may weaken the interaction between excitons, leading to a reduction in the IX splitting. This mechanism may account for the observed decrease in the valley splitting.

The discussion here has been included in **Supplementary Section 13**.

Fig. R13 The valley splitting of the IX as a function of excitation power. The initial increase in splitting with power may be related to valley-dependent exchange interactions, while the subsequent decrease in splitting with power may be due to Coulomb screening.

9. *Can the authors also explicitly explain how both intra and intervalley configuration is possible? Due to charge transfer and H type heterobilayer formation only +K spin down (up) conduction band (WS2) and same spin valence band of WSe2 should have effective electron and hole population. This should always give rise to singlet biexciton formation and “zero” DCP.*

Authors Reply:

We thank the reviewers for this insightful question, which indeed touches upon a central point of our study. As we described in the response to Question 1, the rapid intervalley scattering of interlayer excitons leads to their distribution across both +K and -K valleys, even under selective excitation of the -K valley. When an additional exciton is introduced into the same moiré potential, it can combine with these excitons to form either an intervalley or an intravalley biexciton, depending on the valley configuration of the constituent excitons. For a more detailed explanation, we refer the reviewer to our response to R2Q1 (the first question raised by the reviewer 2).

10. *The non-radiative decay to from interlayer excitonic to biexciton state is obvious at higher optical power and it should affect the total decay time measured by TRPL. How does the exciton decay time change with optical power? The authors should formulate a rate equation considering all these decay components and thereby fit the power dependent TRPL data with the solution of this rate equation. This will help them to calculate the biexciton population at higher power and other non-radiative decay paths.*

Authors Reply:

We thank the reviewers for their highly constructive suggestions. In response to the comments on non-radiative recombination and exciton dynamics, we systematically measured the power-dependent lifetimes of the IX and the IXX and fitted the dynamics using the rate equations. This approach and the related discussion have provided a more comprehensive dataset and solidified the analytical framework of this research, which is detailed below.

We performed the power-dependent TRPL of the IX and the IXX (Fig. R14). In Fig. R14a, we performed a bi-exponential fit for the IX, and found that the fast, slow, and average lifetimes remain unchanged (Fig. R14b). The average lifetime (1 μ s) of IX is well consistent with the observation in other report (*Nat. Phys.* 19, 1286-1292 (2023)). In sharp contrast, the IXX exhibits two orders of magnitude shorter lifetime of about 6 ns at the different powers (Fig. R14c and d). Recent reports (*2D Mater.* 11, 25030

(2024)) show that the biexciton's radiative recombination lifetime exceeds half of the single exciton lifetime ($\tau_{\text{IXX}}/\tau_{\text{IX}} > 0.5$), strongly indicating that the IXX's fast decay here does not originate from any radiative recombination process. Additionally, given the close distance of the two constituent excitons of the biexciton, *i.e.*, approximately 2 nm (*Nat. Mater.* 19, 624-629 (2020)), the exciton-exciton annihilation (EEA) is significantly enhanced. We speculate that the EEA is the main reason for this short lifetime. The biexciton lifetime decreases slightly with the rising powers, probably due to the enhanced EEA resulting from the density changes. Therefore, these power-dependent lifetime measurements suggest that the IX's relaxation is dominated by its own recombination, while the IXX's relaxation is most probably driven by EEA.

Fig. R14 The IX's and the IXX's lifetimes as a function of the excitation power. **a** The time-resolved photoluminescence of IX at different excitation powers, using repetition frequencies of 2 MHz. The black solid line represents the corresponding double exponential fit. **b** Fast, slow, and average lifetimes of the IX obtained from the fitting. **c** The normalized IXX's time-resolved photoluminescence at different excitation powers, using repetition frequencies of 38 MHz. Black arrows indicate a decrease in lifetime of the IXX. **d** IXX's lifetime as a function of excitation power. The lifetime is approximately 6 ns.

Due to the absence of obvious non-radiative recombination for IX and the huge difference in lifetime between the IXX and the IX, the simultaneous fitting of their lifetimes at a unified repetition rate is not feasible, particularly at high repetition rates where the slow decay of IX leads to inaccurate fitting. Consequently, following the reviewers' suggestion, we construct a rate equation model based on the obtained IXX

and IX lifetimes to fit the power-dependent PL intensity. The details are as follows.

Following the reviewers' suggestion, we constructed a rate equation model to describe the IX's and the IXX's dynamics. Guided by the schematic in Fig. R15a and the previous report (*Nat. Mater.* 19, 624-629 (2020)), we established the following rate equations:

$$\frac{dn_1}{dt} = -\frac{n_1}{\tau_{IX}} - gSn_1 + \frac{n_2}{\tau_{IXX}} + gSn_0 \quad (4)$$

$$\frac{dn_2}{dt} = -\frac{n_2}{\tau_{IXX}} + gSn_1 \quad (5)$$

$$\frac{dn_0}{dt} = -gSn_0 + \frac{n_1}{\tau_{IX}} \quad (6)$$

$$n_0 + n_1 + n_2 = n_M \quad (7)$$

$$g = \alpha P \quad (8)$$

Here, n_0 and n_M represent the number of the empty and the total moiré potential wells within the irradiated area by the excitation light, respectively. n_1 and n_2 represent the number of single excitons and biexcitons, respectively (as sketched in Fig. R15a). τ_{IX} and τ_{IXX} are the lifetimes of IX and IXX, respectively. g is the exciton generation rate, where $\alpha \approx 7.5 \times 10^{18} \text{ s}^{-1} \text{ cm}^{-2} \mu\text{W}^{-2}$ (see supplementary Section 4), S is the area of a single unit cell of the moiré superlattice, and thus gS represents the rate of generating an exciton in single unit cell.

Considering that the lifetimes of the IX and the IXX do not change dramatically over the excitation power range ($\tau_{IX} \sim 1 \mu\text{s}$ and $\tau_{IXX} \sim 6 \text{ ns}$), we can investigate their I-P relationship using the rate equations, whose solution is:

$$n_1 = \frac{S\tau_{IX} \cdot g}{1 + S\tau_{IX} \cdot g + S^2\tau_{IX}\tau_{IXX} \cdot g^2} n_M \quad (9)$$

$$n_2 = \frac{S^2\tau_{IX}\tau_{IXX} \cdot g^2}{1 + S\tau_{IX} \cdot g + S^2\tau_{IX}\tau_{IXX} \cdot g^2} n_M \quad (10)$$

Accordingly, we tried to fit the power dependence of the IX and the IXX relaxation using the functions $I_{IX} \sim n_1 = \frac{C_{IX} \cdot P}{1 + A \cdot P + B \cdot P^2}$ and $I_{IXX} \sim n_2 = \frac{C_{IXX} \cdot P^2}{1 + A \cdot P + B \cdot P^2}$, respectively. Where $A = S\tau_{IX}\alpha$, $B = S^2\tau_{IX}\tau_{IXX}\alpha^2$, $C_{IX} = S\tau_{IX}\alpha n_M$, and $C_{IXX} = S^2\tau_{IX}\tau_{IXX}\alpha^2 n_M$. As shown in the Fig. R15b, the fitting result does not match the IX data, although it matches the IXX data. From the expression of the solution, the order of the numerator (P) of n_1 is lower than that of the denominator (P^2). Therefore, n_1 is expected to decrease at high powers. Obviously, this does not conform to the experimental results. Therefore, another factor neglected in this physical picture needs to be considered.

Fig. R15 The rate equation fitting without considering the diffusion. **a** Three-level model of IXX, IX, and empty moiré potential. **b** The rate equation fitting of the power-dependent PL intensity for the IX and the IXX.

Another report indicates (*Nat. Mater.* 24, 527-534 (2025)) that the interlayer excitons undergo notable diffusion as their density remains lower than the moiré density. However, when the exciton density exceeds the moiré density, they become "frozen" as their diffusion is suppressed. Therefore, it is necessary to consider whether the exciton diffusion affects the rate equation. We qualitatively observed the exciton diffusion by keeping the excitation spot size constant and varying the collection spot size. As shown in Fig. R16a, significant difference is observed at an excitation power of 30 μW : the IX intensity is higher when using a larger collection spot size ($D \sim 10 \mu\text{m}$), whereas the IXX intensity becomes higher when using a smaller one ($D \sim 1 \mu\text{m}$). This strongly indicates that the IX and the IXX have different spatial distributions and diffusion behaviors.

For quantitative analysis, we compared the intensity ratios ($I_{(D \sim 10 \mu\text{m})} / I_{(D \sim 1 \mu\text{m})}$) at various excitation powers. The top of Fig. R16b shows the integrated intensity ratio of the full spectrum (both exciton species are included), which significantly increases with the increasing power. This phenomenon may arise from the enhanced diffusion caused by dipole repulsion as the density increases. As illustrated in the bottom of Fig. R16b, the IXX's intensity ratio remains almost constant across the range of powers, whereas the IX's intensity ratio increases remarkably. This suggests that the IXX are primarily confined to a small region around the excitation spot, while the IX can diffuse to areas far from the excitation spot. We attribute this confinement to the short IXX lifetime (Fig. R14d), which curtails the diffusion of the IXX.

Fig. R16 The difference in the IX and the IXX intensity changed with the collection spot size. **a** Comparison of collection intensity between large ($D \sim 10 \mu\text{m}$) and small ($D \sim 1 \mu\text{m}$) spot sizes at the higher and the lower excitation power. **b** The intensity ratio ($I_{(D \sim 10 \mu\text{m})} / I_{(D \sim 1 \mu\text{m})}$) as a function of the excitation power. The top shows the total intensity ratio. Its increase indicates that the exciton diffusion intensifies with the increasing power. The bottom shows the intensity ratios for the IX and the IXX. The ratio of IX increases with excitation power, while the ratio of IXX remains essentially unchanged. This suggests that the biexcitons exhibit minimal diffusion, whereas the IX notably diffuses away from the excitation point.

Therefore, we revise the rate equation by considering the exciton diffusion effect. In this case, the sum of $n_0 + n_1 + n_2$ is no longer equal to the total number of moiré potential wells. Instead, we expand it to the first order of the exciton generation rate, resulting in $n_0 + n_1 + n_2 = n_M + gn_D + \delta(g^2)$, where n_D is the coefficient related to the exciton diffusion. The solution to the revised equations is as follows:

$$n_1 = \frac{Sn_D\tau_{IX} \cdot g^2 + Sn_M\tau_{IX} \cdot g}{1 + S\tau_{IX} \cdot g + S^2\tau_{IX}\tau_{IXX} \cdot g^2} \quad (11)$$

$$n_2 = \frac{S^2\tau_{IX}\tau_{IXX}n_D \cdot g^3 + S^2\tau_{IX}\tau_{IXX}n_M \cdot g^2}{1 + S\tau_{IX} \cdot g + S^2\tau_{IX}\tau_{IXX} \cdot g^2} \quad (12)$$

Considering the exciton diffusion, the order of the numerator (P^2) and denominator (P^2) of n_1 are the same, and the IX intensity no longer decreases at high powers, which is more consistent with the experimental results. We use the simplified functions $I_{IX} \sim n_1 = \frac{C_{IX}P^2 + D_{IX}P}{1 + AP + BP^2}$ and $I_{IXX} \sim n_2 = \frac{C_{IXX}P^3 + D_{IXX}P^2}{1 + AP + BP^2}$ to fit the power dependence of the IX and IXX intensities. The results are shown in Fig. R17, one can see that the equations now fit the experimental results fairly well. The fitting parameters are $A = 4.26 \mu\text{W}^{-1}$, $B = 0.158 \mu\text{W}^{-2}$, $C_{IX} = 8749 \mu\text{W}^{-2}$, $D_{IX} = 1124 \mu\text{W}^{-1}$,

$C_{IXX} = 0 \mu W^{-3}$, $D_{IXX} = 1816 \mu W^{-2}$, respectively. The fitted ratio of $\tau_{IX}/\tau_{IXX} = A^2/B \sim 115$, close to the experimental counterpart $\tau_{IX}/\tau_{IXX} = \frac{1 \mu s}{6 ns} \sim 166$. The fitted area of a single unit cell of the moiré superlattice $S = \frac{A}{\alpha \tau_{IX}} = 57 nm^2$ also agrees well with the result estimated from the moiré period of $7.6 nm$ ($S \sim a_M^2 = 58 nm^2$). Furthermore, one can see that the fitted $\frac{C_{IX}}{D_{IX}} \gg \frac{C_{IXX}}{D_{IXX}} = 0$, contradicting with the expected equal result n_D/n_M (related to diffusion). We attribute this discrepancy to the greatly different diffusion behaviors of the IX and the IXX. As shown in the Fig. R16b, the IX exhibits significant diffusion, while the IXX shows almost no diffusion (n_D of IXX approaches 0), which aligns well with this fitting result.

Fig. R17 The rate equation fitting with considering the exciton diffusion. The gray solid line represents the fitting result.

In summary, the IX primarily undergoes the radiative recombination with a long lifetime of approximately $1 \mu s$. However, in contrast, the enhanced EEA of the IXX greatly shortens its lifetime to $\sim 6 ns$. Furthermore, through considering the impact of exciton diffusion, we fitted the power-dependent TRPL intensity using a rate equation. The fitting results align well with our experimental data. Nevertheless, the presence of IX diffusion makes it rather challenging to obtain the precise exciton densities, which necessitates further in-depth investigation in the future. The discussions here have been included in **Supplementary Section 7 and main text**.

11. *The authors have shown interlayer excitons and biexcitons in the same harmonic well configuration In Figure 2c. The biexciton transition to excitonic state should be depicted between two different harmonic wells as articulated in ref. 2D Mater. 11 (2024) 025030. Also, what are the different radiative decay pathways to excitonic states (harmonic well states) from intra and intervalley biexcitons?*

Authors Reply:

Thank the reviewers for the valuable suggestions and questions. We are sorry for the misunderstanding, which may arise from the insufficient clarity in the original figure captions. In Fig. 2c in the manuscript, we depict the distribution of interlayer excitons and interlayer biexcitons across different moiré potential wells in real space. Specifically, the single excitons, intervalley biexcitons, and intravalley biexcitons are situated in the harmonic oscillator wells at spatially distinct locations. Moreover, the schematic also accounts for the energy difference between the intravalley/intervalley biexcitons and the single excitons, as referenced in prior works such as Fig. 3c of *2D Mater.* 11, 025030 (2024). Accordingly, the dashed lines are used to indicate the energy differences between IXX (X_+X_- , X_-X_-) and IX.

Regarding the relaxation of IXX, the literature (*2D Mater.* 11, 025030 (2024)) indicates that the interlayer excitons might occupy the excited levels, suggesting a possible relaxation route of the biexcitons into these states. However, the calculation of this literature shows that the biexciton luminescence is primarily dominated by its transition to the ground state of the single exciton. This study finds no evidence of an IXX relaxation process to excited states. If such a process existed, it would produce a series of alternately polarized IXX peaks in the PL spectra. Similarly, no emissions from IX excited states are observed, which would otherwise appear as multiple, alternately polarized IX peaks under low excitation powers (*Nature.* 567, 71-75 (2019)). This behavior matches well with the other report (*Nat. Phys.* 19, 1286-1292 (2023); *Nat. Mater.* 19, 624-629 (2020)), where only IXX-to-ground-state-IX relaxation is detected.

Finally, we demonstrate the relaxation process from IXX to IX in the supplementary section 10 and Fig. R18. The intravalley biexciton X_-X_- and intervalley biexciton X_+X_- emit polarized and unpolarized light, respectively, as they relax to the IX ground state. Additionally, to avoid misinterpretation, we have used the yellow, the gray and the red color to mark the harmonic oscillator wells of IX, X_+X_- , and X_-X_- , respectively. we have added clearer descriptions in the main text and figure legend of Fig. 2c.

The discussion here has been included in **Supplementary Section 10**.

Fig. R18 The relaxation process of biexciton to single exciton. An intravalley biexciton emits circularly polarized light, while an intervalley biexciton emits unpolarized light. Furthermore, the relaxation of biexciton states is primarily dominated by their transitions to the ground state exciton, with no observable experimental evidence of relaxation to the exciton's excited states. The red, gray, and yellow harmonic oscillator wells represent the intravalley IXX, intervalley IXX, and IX, respectively. Δ_{IXX} represents the fine structure splitting of the IXX.

12. Minor comment: Supplementary section 6 caption is repeated twice while the supplementary section 7 header is absent. Please revise accordingly.

Authors Reply:

Thank the reviewers for their comment. The titles of Supplementary section 6 and Supplementary section 7, which were previously “The TRPL of the IXX” and “The TRPL of the IX” respectively, were misleading. We have revised them to “The TRPL of the interlayer biexciton” and “The TRPL of the interlayer single exciton”.

Again, we deeply appreciate the reviewers' thoughtful feedback and valuable suggestions. All the points raised have been addressed, the corresponding revisions have been made: the interlayer exciton formation processes related to layers (Fig. R5), the power dependence FWHM of the IX (Fig. R6), biexciton cascade emission (Fig. R7), The FWHM of interlayer exciton and the emission of intralayer exciton in monolayer and heterostructure regions (Fig. R8), the IX' and IXX' emission peak positions as a function of the excitation power (Fig. R9), magnetic field dependent of DCP for the IX and the IXX at 4 K (Fig. R10), exchange interaction measurement at different excitation powers (Fig. R11), exchange interaction measurement at different temperatures (Fig. R12), The valley splitting of the IX as a function of excitation power (Fig. R13), the lifetime of IX and IXX as a function of the excitation power (Fig. R14), the rate equation fitting without/with diffusion (Fig. R15 and Fig. R17) and the relaxation process of biexciton to single exciton (Fig. R18) as well as the related discussions have been added to the revised main text and supplementary information. We believe these substantial improvements provide a more comprehensive understanding of this work, which we hope to meet the reviewers' expectations.

Finally, we are truly grateful to the reviewers for their valuable comments and constructive suggestions. All the issues raised have been carefully considered and addressed, all the revisions are fully reflected in the updated manuscript and the supplementary information. We believe these substantial improvements provide a more comprehensive understanding and strengthen the importance of this work, which we hope to meet the reviewers' expectations.

REVIEWERS' COMMENTS

Reviewer #1 (Remarks to the Author):

The authors have appropriately addressed most of my concerns. In my opinion, the revised manuscript upholds the scope and requirements of the journal and should be published.

Reply:

We thank reviewer for his/her review and support for the publication of our work.

Reviewer #2 (Remarks to the Author):

We would first like to thank the authors for their detailed responses to the most of our concerns or doubts, as well as undertaking additional experimental works. The new data solve almost all our queries effectively and improves the quality of the manuscript. We only have a few minor comments remaining; nonetheless, the manuscript is almost ready to be accepted in Nature Communications.:

Authors Reply:

We sincerely thank the reviewers for their positive assessment that our manuscript is “almost ready to be accepted in Nature Communications”. We also thank their thoughtful and valuable feedback, which has greatly helped us to clarify and strengthen the discussion in our manuscript. In response, we have further discussed the relevant physical picture and made corresponding revisions, which have enhanced the rigor of our work.

Remarks and Questions:

- 1. In response to comment #2, the authors attribute the slight increase in biexcitonic power law with temperature to the exciton delocalization effect from shallow non-radiative potentials. Is this delocalization related to defect sites? Throughout the manuscript, the potentials have primarily been ascribed to moiré potentials, which appear to facilitate radiative recombination. Consequently, the reference to “non-radiative” in Supplementary 6 creates some ambiguity.*

Authors Reply:

We thank the reviewer for the valuable suggestion. Shallow non-radiative potential wells are likely associated with defect states, are commonly observed in transition metal dichalcogenides, as reported in *Nat. Commun.* 14, 6910 (2023). Consequently, to describe more clearly and accurately, we have revised the expression of “We attribute this behavior to the exciton delocalization effect from shallow non-radiative potential” to “We attribute this behavior to the exciton delocalization effect from shallow non-radiative potential, which is might related to the defect sites in the sample” in **the supplementary section 6**.

2. *The authors propose that exciton-exciton annihilation in IXX accounts for the rapid total lifetime observed in TRPL. However, unlike IX, where multiple light cones exist between two parallel bands (IXX and IX band), this phenomenon does not apply to IXX. We suggest that the faster decay of IXX relative to IX results from non-radiative processes, given the increased Bohr radius arising from strong dipolar repulsion between two excitons.*

Authors Reply:

We sincerely thank the reviewers for raising the question regarding the origin of the short lifetime of IXX. Below, we provide a detailed discussion of its potential causes.

1. As suggested by the reviewer, we admit that the faster decay of IXX might result from the enhanced non-radiative processes, because the strong dipolar repulsion between two excitons in the IXX can lead to its dissociation. However, considering that biexcitons can still exist at high temperatures of 120K, we believe that the dissociation effect from dipolar repulsion is not large enough to account for the nearly two orders of magnitude reduction in the exciton lifetime. Therefore, we suspect that there might be other mechanisms as well for such fast lifetime of IXX.
2. It has been reported that, also in H-WSe₂/WS₂ moiré superlattice, a biexciton has a fast lifetime of approximately 4.3 ns, which is attributed to exciton-exciton Auger effect (EEA) (*Nat. Mater.* 24, 527-534 (2025)). Inspired by this result, we also infer that the fast lifetime of IXX in our study mainly originates from the EEA for the additional reasons: First, the two excitons that form a biexciton are very close each other, which makes Auger effect particularly prone to occurring. Second, in the TMDs, the EEA rate can be described by $\gamma_A = K_A n_{\text{eff}}$, where K_A is the EEA coefficient and n_{eff} is the local exciton density. The EEA rate of biexcitons can be estimated using the local exciton density: $\gamma_A = \frac{2K_A}{\pi r^2}$, where r is exciton radius of IXX. According to the report

(*Nano Lett.* 24, 2773-2781 (2024)), the EEA coefficient of interlayer excitons is approximately $10^{-5} \text{cm}^2 \text{s}^{-1}$. Substituting $r = 2 \text{ nm}$, we can estimate the EEA rate to be approximately $1.6 \times 10^8 \text{s}^{-1}$. The rate corresponds to a lifetime of 6 ns, which is very close with our experimental results shown in Fig.S9d.

In summary, we believe that, in the current study, the fast lifetime of biexcitons is mainly due to the EEA process, while the non-radiative recombination may also have contribution. The detailed underlying mechanism still needs to be clarified theoretically. This discussion has been included in **the Supplementary section 7**